# BackPACK: Packing more into Backprop

**Felix Dangel**\*
University of Tuebingen
fdangel@tue.mpg.de

**Frederik Kunstner**\*
University of Tuebingen
kunstner@cs.ubc.ca

**Philipp Hennig**
University of Tuebingen and MPI for Intelligent Systems, Tuebingen
ph@tue.mpg.de

## Abstract

Automatic differentiation frameworks are optimized for exactly one thing: computing the average mini-batch gradient. Yet, other quantities such as the variance of the mini-batch gradients or many approximations to the Hessian can, *in theory*, be computed efficiently, and at the same time as the gradient. While these quantities are of great interest to researchers and practitioners, current deep-learning software does not support their automatic calculation. Manually implementing them is burdensome, inefficient if done naïvely, and the resulting code is rarely shared. This hampers progress in deep learning, and unnecessarily narrows research to focus on gradient descent and its variants; it also complicates replication studies and comparisons between newly developed methods that require those quantities, to the point of impossibility. To address this problem, we introduce BackPACK[1], an efficient framework built on top of PyTorch, that extends the backpropagation algorithm to extract additional information from first- and second-order derivatives. Its capabilities are illustrated by benchmark reports for computing additional quantities on deep neural networks, and an example application by testing several recent curvature approximations for optimization.

## 1 Introduction

The success of deep learning and the applications it fuels can be traced to the popularization of automatic differentiation frameworks. Packages like TensorFlow (Abadi et al., 2016), Chainer (Tokui et al., 2015), MXNet (Chen et al., 2015), and PyTorch (Paszke et al., 2019) provide efficient implementations of parallel, GPU-based gradient computations to a wide range of users, with elegant syntactic sugar.

However, this specialization also has its shortcomings: it assumes the user only wants to compute gradients or, more precisely, the average of gradients across a mini-batch of examples. Other quantities can also be computed with automatic differentiation at a comparable cost or minimal overhead to the gradient backpropagation pass; for example, approximate second-order information or the variance of gradients within the batch. These quantities are valuable to understand the geometry of deep neural networks, for the identification of free parameters, and to push the development of more efficient optimization algorithms. But researchers who want to investigate their use face a chicken-and-egg problem: automatic differentiation tools required to go beyond standard gradient methods are not available, but there is no incentive for their implementation in existing deep-learning software as long as no large portion of the users need it.

Second-order methods for deep learning have been continuously investigated for decades (e.g., Becker & Le Cun, 1989; Amari, 1998; Bordes et al., 2009; Martens & Grosse, 2015). But still, the standard optimizers used in deep learning remain some variant of stochastic gradient descent (SGD); more complex methods have not found wide-spread, practical use. This is in stark contrast to domains like convex optimization and generalized linear models, where second-order methods are

---

\*Equal contributions
[1]https://f-dangel.github.io/backpack/

the default. There may of course be good scientific reasons for this difference; maybe second-order methods do not work well in the (non-convex, stochastic) setting of deep learning. And the computational cost associated with the high dimensionality of deep models may offset their benefits. Whether these are the case remains somewhat unclear though, because a much more direct road-block is that these methods are so complex to implement that few practitioners ever try them out.

Recent approximate second-order methods such as KFAC (Martens & Grosse, 2015) show promising results, even on hard deep learning problems (Tsuji et al., 2019). Their approach, based on the earlier work of Schraudolph (2002), uses the structure of the network to compute approximate second-order information in a way that is similar to gradient backpropagation. This work sparked a new line of research to improve the second-order approximation (Grosse & Martens, 2016; Botev et al., 2017; Martens et al., 2018; George et al., 2018). However, all of these methods require low-level applications of automatic differentiation to compute quantities other than the averaged gradient. It is a daunting task to implement them from scratch. Unless users spend significant time familiarizing themselves with the internals of their software tools, the resulting implementation is often inefficient, which also puts the original usability advantage of those packages into question. Even motivated researchers trying to develop new methods, who need not be expert software developers, face this problem. They often end up with methods that cannot compete in runtime, not necessarily because the method is inherently bad, but because the implementation is not efficient. New methods are also frequently not compared to their predecessors and competitors because they are so hard to reproduce. Authors do not want to represent the competition in an unfair light caused by a bad implementation.

Another example is offered by a recent string of research to adapt to the *stochasticity* induced by mini-batch sampling. An empirical estimate of the (marginal) variance of the gradients within the batch has been found to be theoretically and practically useful for adapting hyperparameters like learning rates (Mahsereci & Hennig, 2017) and batch sizes (Balles et al., 2017), or regularize first-order optimization (Le Roux et al., 2007; Balles & Hennig, 2018; Katharopoulos & Fleuret, 2018). To get such a variance estimate, one simply has to square, then sum, the individual gradients after the backpropagation, but before they are aggregated to form the average gradient. Doing so should have negligible cost *in principle*, but is programmatically challenging in the standard packages.

Members of the community have repeatedly asked for such features[2] but the established automatic differentiation frameworks have yet to address such requests, as their focus has been—rightly—on improving their technical backbone. Features like those outlined above are not generally defined for arbitrary functions, but rather emerge from the specific structure of machine learning applications. General automatic differentiation frameworks can not be expected to serve such specialist needs. This does not mean, however, that it is impossible to efficiently realize such features within these frameworks: In essence, backpropagation is a technique to compute multiplications with Jacobians. Methods to extract second-order information (Mizutani & Dreyfus, 2008) or individual gradients from a mini-batch (Goodfellow, 2015) have been known to a small group of specialists; they are just rarely discussed or implemented.

## 1.1 OUR CONTRIBUTION

To address this need for a specialized framework focused on machine learning, we propose a framework for the implementation of generalized backpropagation to compute additional quantities. The structure is based on the conceptual work of Dangel et al. (2019) for modular backpropagation. This framework can be built on top of existing graph-based backpropagation modules; we provide an implementation on top of PYTORCH, coined BACKPACK, available at



`https://f-dangel.github.io/backpack/`.



The initial release supports efficient computation of individual gradients from a mini-batch, their $\ell_2$ norm, an estimate of the variance, as well as diagonal and Kronecker factorizations of the generalized Gauss-Newton (GGN) matrix (see Tab. 1 for a feature overview). The library was designed to be minimally verbose to the user, easy to use (see Fig. 1), and to have low overhead (see §3). While other researchers are aiming to improve the flexibility of automatic differentiation systems (Innes, 2018a;b; Bradbury et al., 2018), our goal with this package is to provide access to quantities that are only byproducts of the backpropagation pass, rather than gradients themselves.

---

[2] See, e.g., the Github issues `github.com/pytorch/pytorch/issues/1407, 7786, 8897` and forum discussions `discuss.pytorch.org/t/1433, 8405, 15270, 17204, 19350, 24955`.

**Computing the gradient with PYTORCH ...**

```
X, y      = load_mnist_data()
model     = Linear(784, 10)
lossfunc  = CrossEntropyLoss()

loss      = lossfunc(model(X), y)

loss.backward()

for param in model.parameters():
    print(param.grad)
```

**... and the variance with BACKPACK**

```
X, y      = load_mnist_data()
model     = extend(Linear(784, 10))
lossfunc  = extend(CrossEntropyLoss())

loss      = lossfunc(model(X), y)
with backpack(Variance()):
    loss.backward()

for param in model.parameters():
    print(param.grad)
    print(param.var)
```

Figure 1: BACKPACK integrates with PYTORCH to seamlessly extract more information from the backward pass. Instead of the variance (or alongside it, in the same pass), BACKPACK can compute individual gradients in the mini-batch, their $\ell_2$ norm and $2^{\text{nd}}$ moment. It can also compute curvature approximations like diagonal or Kronecker factorizations of the GGN such as KFAC, KFLR & KFRA.

To illustrate the capabilities of BACKPACK, we use it to implement preconditioned gradient descent optimizers with diagonal approximations of the GGN and recent Kronecker factorizations KFAC (Martens & Grosse, 2015), KFLR, and KFRA (Botev et al., 2017). Our results show that the curvature approximations based on Monte-Carlo (MC) estimates of the GGN, the approach used by KFAC, give similar progress per iteration to their more accurate counterparts, but being much cheaper to compute. While the naïve update rule we implement does not surpass first-order baselines such as SGD with momentum and Adam (Kingma & Ba, 2015), its implementation with various curvature approximations is made straightforward.

## 2 THEORY AND IMPLEMENTATION

We will distinguish between quantities that can be computed from information already present during a traditional backward pass (which we suggestively call *first-order extensions*), and quantities that need additional information (termed *second-order extensions*). The former group contains additional statistics such as the variance of the gradients within the mini-batch or the $\ell_2$ norm of the gradient for each sample. Those can be computed with minimal overhead during the backprop pass. The latter class contains approximations of second-order information, like the diagonal or Kronecker factorization of the generalized Gauss-Newton (GGN) matrix, which require the propagation of additional information through the graph. We will present those two classes separately:

| **First-order extensions** | **Second-order extensions** |
|---|---|
| Extract more from the standard backward pass. | Propagate new information along the graph. |
| – Individual gradients from a mini-batch | – Diagonal of the GGN and the Hessian |
| – $\ell_2$ norm of the individual gradients | – KFAC (Martens & Grosse, 2015) |
| – Diagonal covariance and $2^{\text{nd}}$ moment | – KFRA and KFLR (Botev et al., 2017) |

These quantities are only defined, or reasonable to compute, for a subset of models: The concept of individual gradients for each sample in a mini-batch or the estimate of the variance requires the loss for each sample to be independent. While such functions are common in machine learning, not all neural networks fit into this category. For example, if the network uses Batch Normalization (Ioffe & Szegedy, 2015), the individual gradients in a mini-batch are correlated. Then, the variance is not meaningful anymore, and computing the individual contribution of a sample to the mini-batch gradient or the GGN becomes prohibitive. For those reasons, and to limit the scope of the project for version 1.0, BACKPACK currently restricts the type of models it accepts. The supported models are traditional feed-forward networks that can be expressed as a *sequence of modules*, for example a sequence of convolutional, pooling, linear and activation layers. Recurrent networks like LSTMs (Hochreiter & Schmidhuber, 1997) or residual networks (He et al., 2016) are not yet supported, but the framework can be extended to cover them.

We assume a sequential model $f : \Theta \times \mathbb{X} \to \mathbb{Y}$ and a dataset of $N$ samples $(\boldsymbol{x}_n, \boldsymbol{y}_n) \in \mathbb{X} \times \mathbb{Y}$ with $n = 1, \ldots, N$. The model maps each sample $\boldsymbol{x}_n$ to a prediction $\hat{\boldsymbol{y}}_n$ using some parameters $\boldsymbol{\theta} \in \Theta$. The predictions are evaluated with a loss function $\ell : \mathbb{Y} \times \mathbb{Y} \to \mathbb{R}$, for example the cross-entropy,

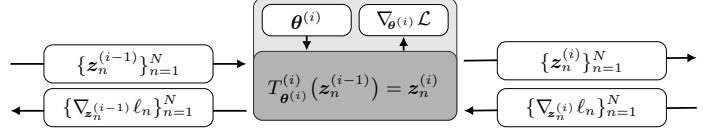

Figure 2: Schematic representation of the standard backpropagation pass for module $i$ with $N$ samples.

which compares them to the ground truth $\boldsymbol{y}_n$. This leads to the objective function $\mathcal{L} : \Theta \to \mathbb{R}$,

$$\mathcal{L}(\boldsymbol{\theta}) = \tfrac{1}{N} \sum_{n=1}^{N} \ell(f(\boldsymbol{\theta}, \boldsymbol{x}_n), \boldsymbol{y}_n). \tag{1}$$

As a shorthand, we will use $\ell_n(\boldsymbol{\theta}) = \ell(f(\boldsymbol{\theta}, \boldsymbol{x}_n), \boldsymbol{y}_n)$ for the loss and $f_n(\boldsymbol{\theta}) = f(\boldsymbol{\theta}, \boldsymbol{x}_n)$ for the model output of individual samples. Our goal is to provide more information about the derivatives of $\{\ell_n\}_{n=1}^{N}$ with respect to the parameters $\boldsymbol{\theta}$ of the model $f$.

## 2.1 PRIMER ON BACKPROPAGATION

Machine learning libraries with integrated automatic differentiation use the modular structure of $f_n(\boldsymbol{\theta})$ to compute derivatives (see Baydin et al. (2018) for an overview). If $f_n$ is a sequence of $L$ transformations, it can be expressed as

$$f_n(\boldsymbol{\theta}) = T_{\boldsymbol{\theta}^{(L)}}^{(L)} \circ \ldots \circ T_{\boldsymbol{\theta}^{(1)}}^{(1)}(\boldsymbol{x}_n), \tag{2}$$

where $T_{\boldsymbol{\theta}^{(i)}}^{(i)}$ is the $i$th transformation with parameters $\boldsymbol{\theta}^{(i)}$, such that $\boldsymbol{\theta} = [\boldsymbol{\theta}^{(1)}, \ldots, \boldsymbol{\theta}^{(L)}]$. The loss function can also be seen as another transformation, appended to the network. Let $\boldsymbol{z}_n^{(i-1)}, \boldsymbol{z}_n^{(i)}$ denote the input and output of the operation $T_{\boldsymbol{\theta}^{(i)}}^{(i)}$ for sample $n$, such that $\boldsymbol{z}_n^{(0)}$ is the original data and $\boldsymbol{z}_n^{(1)}, \cdots, \boldsymbol{z}_n^{(L)}$ represent the transformed output of each layer, leading to the computation graph

$$\boldsymbol{z}_n^{(0)} \xrightarrow{T_{\boldsymbol{\theta}^{(1)}}^{(1)}(\boldsymbol{z}_n^{(0)})} \boldsymbol{z}_n^{(1)} \xrightarrow{T_{\boldsymbol{\theta}^{(2)}}^{(2)}(\boldsymbol{z}_n^{(1)})} \ldots \xrightarrow{T_{\boldsymbol{\theta}^{(L)}}^{(L)}(\boldsymbol{z}_n^{(L-1)})} \boldsymbol{z}^{(L)} \xrightarrow{\ell(\boldsymbol{z}_n^{(L)}, \boldsymbol{y}_n)} \ell_n(\boldsymbol{\theta}).$$

To compute the gradient of $\ell_n$ with respect to the $\boldsymbol{\theta}^{(i)}$, one can repeatedly apply the chain rule,

$$\begin{aligned}
\nabla_{\boldsymbol{\theta}^{(i)}} \ell(\boldsymbol{\theta}) &= (\mathrm{J}_{\boldsymbol{\theta}^{(i)}} \boldsymbol{z}_n^{(i)})^\top (\mathrm{J}_{\boldsymbol{z}_n^{(i)}} \boldsymbol{z}_n^{(i+1)})^\top \ldots (\mathrm{J}_{\boldsymbol{z}_n^{(L-1)}} \boldsymbol{z}_n^{(L)})^\top (\nabla_{\boldsymbol{z}_n^{(L)}} \ell_n(\boldsymbol{\theta})) \\
&= (\mathrm{J}_{\boldsymbol{\theta}^{(i)}} \boldsymbol{z}_n^{(i)})^\top (\nabla_{\boldsymbol{z}^{(i)}} \ell_n(\boldsymbol{\theta})),
\end{aligned} \tag{3}$$

where $\mathrm{J}_{\boldsymbol{a}} \boldsymbol{b}$ is the Jacobian of $\boldsymbol{b}$ with respect to $\boldsymbol{a}$, $[\mathrm{J}_{\boldsymbol{a}} \boldsymbol{b}]_{ij} = \partial [\boldsymbol{b}]_i / \partial [\boldsymbol{a}]_j$. A similar expression exists for the module inputs $\boldsymbol{z}_n^{(i-1)}$: $\nabla_{\boldsymbol{z}_n^{(i-1)}} \ell_n(\boldsymbol{\theta}) = (\mathrm{J}_{\boldsymbol{z}_n^{(i-1)}} \boldsymbol{z}_n^{(i)})^\top (\nabla_{\boldsymbol{z}_n^{(i)}} \ell_n(\boldsymbol{\theta}))$. This recursive structure makes it possible to extract the gradient by propagating the gradient of the loss. In the backpropagation algorithm, a module $i$ receives the loss gradient with respect to its output, $\nabla_{\boldsymbol{z}_n^{(i)}} \ell_n(\boldsymbol{\theta})$. It then extracts the gradient with respect to its parameters and inputs, $\nabla_{\boldsymbol{\theta}^{(i)}} \ell_n(\boldsymbol{\theta})$ and $\nabla_{\boldsymbol{z}_n^{(i-1)}} \ell_n(\boldsymbol{\theta})$, according to Eq. 3. The gradient with respect to its input is sent further down the graph. This process, illustrated in Fig. 2, is repeated for each transformation until all gradients are computed. To implement backpropagation, each module only needs to know how to multiply with its Jacobians.

For second-order quantities, we rely on the work of Mizutani & Dreyfus (2008) and Dangel et al. (2019), who showed that a scheme similar to Eq. 3 exists for the *block-diagonal* of the Hessian. A block with respect to the parameters of a module, $\nabla_{\boldsymbol{\theta}^{(i)}}^2 \ell_n(\boldsymbol{\theta})$, can be obtained by the recursion

$$\nabla_{\boldsymbol{\theta}^{(i)}}^2 \ell_n(\boldsymbol{\theta}) = (\mathrm{J}_{\boldsymbol{\theta}^{(i)}} \boldsymbol{z}_n^{(i)})^\top (\nabla_{\boldsymbol{z}_n^{(i)}}^2 \ell_n(\boldsymbol{\theta}))(\mathrm{J}_{\boldsymbol{\theta}^{(i)}} \boldsymbol{z}_n^{(i)}) + \sum_j \left( \nabla_{\boldsymbol{\theta}^{(i)}}^2 [\boldsymbol{z}_n^{(i)}]_j \right) \left[ \nabla_{\boldsymbol{z}_n^{(i)}} \ell_n(\boldsymbol{\theta}) \right]_j, \tag{4}$$

and a similar relation holds for the Hessian with respect to each module's output, $\nabla_{\boldsymbol{z}_n^{(i)}}^2 \ell_n(\boldsymbol{\theta})$.

Both backpropagation schemes of Eq. 3 and Eq. 4 hinge on the multiplication by Jacobians to both vectors and matrices. However, the design of automatic differentiation limits the application of Jacobians to vectors only. This prohibits the exploitation of vectorization in the matrix case, which is needed for second-order information. The lacking flexibility of Jacobians is one motivation for our work. Since all quantities needed to compute statistics of the derivatives are already computed during the backward pass, another motivation is to provide access to them at minor overhead.

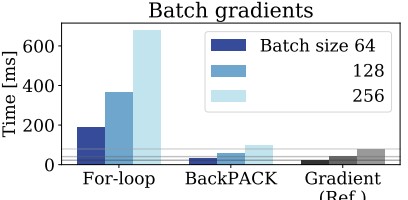

Figure 3: Computing individual gradients in a batch using a for-loop (i.e. one individual forward and backward pass per sample) or using vectorized operations with BACK-PACK. The plot shows computation time, comparing to a traditional gradient computation, on the 3C3D network (See §4) for the CIFAR-10 dataset (Schneider et al., 2019).

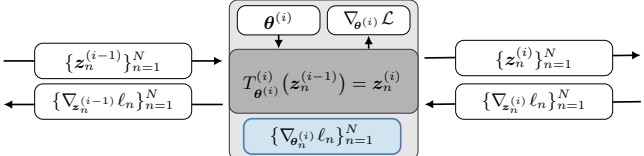

Figure 4: Schematic representation of the individual gradients' extraction in addition to the standard backward pass at the $i$th module for $N$ samples.

## 2.2 FIRST ORDER EXTENSIONS

As the principal first-order extension, consider the computation of the *individual* gradients in a batch of size $N$. These individual gradients are implicitly computed during a traditional backward pass because the batch gradient is their sum, but they are not directly accessible. The naïve way to compute $N$ individual gradients is to do $N$ separate forward and backward passes, This (inefficiently) replaces every matrix-matrix multiplications by $N$ matrix-vector multiplications. BACKPACK's approach batches computations to obtain large efficiency gains, as illustrated by Fig. 3.

As the quantities necessary to compute the individual gradients are already propagated through the computation graph, we can reuse them by inserting code in the standard backward pass. With access to this information, before it is cleared for memory efficiency, BACKPACK computes the Jacobian-multiplications for each sample

$$\{\nabla_{\boldsymbol{\theta}^{(i)}}\ell_n(\boldsymbol{\theta})\}_{n=1}^{N} = \{[\mathrm{J}_{\boldsymbol{\theta}^{(i)}}\boldsymbol{z}_n^{(i)}]^{\top}\nabla_{\boldsymbol{z}_n^{(i)}}\ell_n(\boldsymbol{\theta})\}_{n=1}^{N}\,, \tag{5}$$

without summing the result—see Fig. 4 for a schematic representation. This duplicates some of the computation performed by the backpropagation, as the Jacobian is applied twice (once by PYTORCH and BACKPACK with and without summation over the samples, respectively). However, the associated overhead is small compared to the for-loop approach: The major computational cost arises from the propagation of information required for each layer, rather than the formation of the gradient *within* each layer.

This scheme for individual gradient computation is the basis for all first-order extensions. In this direct form, however, it is expensive in memory: if the model is $D$-dimensional, storing $\mathcal{O}(ND)$ elements is prohibitive for large batches. For the variance, $2^{\text{nd}}$ moment and $\ell_2$ norm, BACKPACK takes advantage of the Jacobian's structure to directly compute them without forming the individual gradient, reducing memory overhead. See Appendix A.1 for details.

## 2.3 SECOND-ORDER EXTENSIONS

Second-order extensions require propagation of more information through the graph. As an example, we will focus on the generalized Gauss-Newton (GGN) matrix (Schraudolph, 2002). It is guaranteed to be positive semi-definite and is a reasonable approximation of the Hessian near the minimum, which motivates its use in approximate second-order methods. For popular loss functions, it coincides with the Fisher information matrix used in natural gradient methods (Amari, 1998); for a more in depth discussion of the equivalence, see the reviews of Martens (2014) and Kunstner et al. (2019). For an objective function that can be written as the composition of a loss function $\ell$ and a model $f$, such as Eq. 1, the GGN of $\frac{1}{N}\sum_n \ell(f(\boldsymbol{\theta},\boldsymbol{x}_n),\boldsymbol{y}_n)$ is

$$G(\boldsymbol{\theta}) = \frac{1}{N}\sum_n \left[\mathrm{J}_{\boldsymbol{\theta}}f(\boldsymbol{\theta},\boldsymbol{x}_n)\right]^{\top}\nabla_f^2\,\ell(f(\boldsymbol{\theta},\boldsymbol{x}_n),\boldsymbol{y}_n)\left[\mathrm{J}_{\boldsymbol{\theta}}f(\boldsymbol{\theta},\boldsymbol{x}_n)\right]. \tag{6}$$

The full matrix is too large to compute and store. Current approaches focus on its diagonal blocks, where each block corresponds to a layer in the network. Every block itself is further approximated,

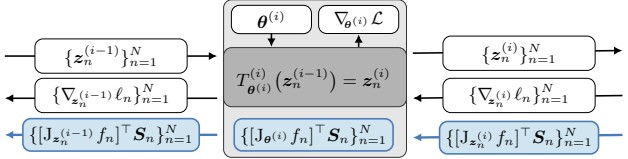

Figure 5: Schematic of the additional backward pass to compute a symmetric factorization of the GGN,
$$G(\boldsymbol{\theta}) = \sum_n [\mathrm{J}_{\boldsymbol{\theta}} f_n]^\top \boldsymbol{S}_n \boldsymbol{S}_n^\top [\mathrm{J}_{\boldsymbol{\theta}} f_n]$$
alongside the gradient at the $i$th module, for $N$ samples.

for example using a Kronecker factorization. The approach used by BACKPACK for their computation is a refinement of the *Hessian Backpropagation equations* of Dangel et al. (2019). It relies on two insights: Firstly, the computational bottleneck in the computation of the GGN is the multiplication with the Jacobian of the network, $\mathrm{J}_{\boldsymbol{\theta}} f_n$, while the Hessian of the loss with respect to the output of the network is easy to compute for most popular loss functions. Secondly, it is not necessary to compute and store each of the $N$ $[D \times D]$ matrices for a network with $D$ parameters, as Eq. 6 is a quadratic expression. Given a symmetric factorization $\boldsymbol{S}_n$ of the Hessian, $\boldsymbol{S}_n \boldsymbol{S}_n^\top = \nabla_f^2 \, \ell(f(\boldsymbol{\theta}, \boldsymbol{x}_n), \boldsymbol{y}_n)$, it is sufficient to compute $[\mathrm{J}_{\boldsymbol{\theta}} f_n]^\top \boldsymbol{S}_n$ and square the result. A network output is typically small compared to its inner layers; networks on CIFAR-100 need $C = 100$ class outputs but could use convolutional layers with more than 100,000 parameters.

The factorization leads to a $[D \times C]$ matrix, which makes it possible to efficiently compute GGN block diagonals. Also, the computation is very similar to that of a gradient, which computes $[\mathrm{J}_{\boldsymbol{\theta}} f_n]^\top \nabla_{f_n} \ell_n$. A module $T_{\boldsymbol{\theta}^{(i)}}^{(i)}$ receives the symmetric factorization of the GGN with respect to its output, $\boldsymbol{z}_n^{(i)}$, and multiplies it with the Jacobians with respect to the parameters $\boldsymbol{\theta}^{(i)}$ and inputs $\boldsymbol{z}_n^{(i-1)}$ to produce a symmetric factorization of the GGN with respect to the parameters and inputs, as shown in Fig. 5.

This propagation serves as the basis of the second-order extensions. If the full symmetric factorization is not wanted, for memory reasons, it is possible to extract more specific information such as the diagonal. If $\boldsymbol{B}$ is the symmetric factorization for a GGN block, the diagonal can be computed as $[\boldsymbol{B}\boldsymbol{B}^\top]_{ii} = \sum_j [\boldsymbol{B}]_{ij}^2$, where $[\cdot]_{ij}$ denotes the element in the $i$th row and $j$th column.

This framework can be used to extract the main Kronecker factorizations of the GGN, KFAC and KFLR, which we extend to convolution using the approach of Grosse & Martens (2016). The important difference between the two methods is the initial matrix factorization $\boldsymbol{S}_n$. Using a full symmetric factorization of the initial Hessian, $\boldsymbol{S}_n \boldsymbol{S}_n^\top = \nabla_{f_n}^2 \ell_n$, yields the KFLR approximation. KFAC uses an MC-approximation by sampling a vector $\boldsymbol{s}_n$ such that $\mathbb{E}_{\boldsymbol{s}_n}[\boldsymbol{s}_n \boldsymbol{s}_n^\top] = \nabla_{f_n}^2 \ell_n$. KFLR is therefore more precise but more expensive than KFAC, especially for networks with high-dimensional outputs, which is reflected in our benchmark on CIFAR-100 in Section 3. The technical details on how Kronecker factors are extracted and information is propagated for second-order BACKPACK extensions are documented in Appendix A.2.

## 3 EVALUATION AND BENCHMARKS

We benchmark the overhead of BACKPACK on the CIFAR-10 and CIFAR-100 datasets, using the 3C3D network[3] provided by DEEPOBS (Schneider et al., 2019) and the ALL-CNN-C[4] network of Springenberg et al. (2015). The results are shown in Fig. 6.

For first-order extensions, the computation of individual gradients from a mini-batch adds noticeable overhead due to the additional memory requirements of storing them. But more specific quantities such as the $\ell_2$ norm, 2nd moment and variance can be extracted efficiently. Regarding second-order extensions, the computation of the GGN can be expensive for networks with large outputs like CIFAR-100, regardless of the approximation being diagonal of Kronecker-factored. Thankfully, the MC approximation used by KFAC, which we also implement for a diagonal approximation, can be computed at minimal overhead—much less than two backward passes. This last point is encouraging, as our optimization experiment in Section 4 suggest that this approximation is reasonably accurate.

---

[3]3C3D is a sequence of 3 convolutions and 3 dense linear layers with 895,210 parameters.

[4]ALL-CNN-C is a sequence of 9 convolutions with 1,387,108 parameters.

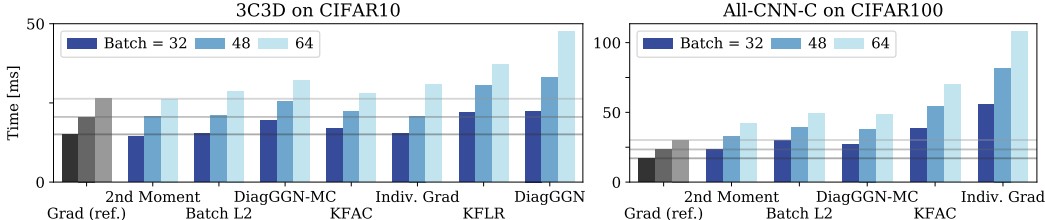

Figure 6: Overhead benchmark for computing the gradient *and* first- or second-order extensions on real networks, compared to just the gradient. Most quantities add little overhead. KFLR and DiagGGN propagate $100\times$ more information than KFAC and DiagGGN-MC on CIFAR-100 and are two orders of magnitude slower. We report benchmarks on those, and the Hessian's diagonal, in Appendix B.

## 4 EXPERIMENTS

To illustrate the utility of BACKPACK, we implement preconditioned gradient descent optimizers using diagonal and Kronecker approximations of the GGN. To our knowledge, and despite their apparent simplicity, results using diagonal approximations or the naïve damping update rule we chose have not been reported in publications so far. However, this section is not meant to introduce a bona-fide new optimizer. Our goal is to show that BACKPACK can enable research of this kind. The update rule we implement uses a curvature matrix $\boldsymbol{G}(\boldsymbol{\theta}_t^{(i)})$, which could be a diagonal or Kronecker factorization of the GGN blocks, and a damping parameter $\lambda$ to precondition the gradient:

$$\boldsymbol{\theta}_{t+1}^{(i)} = \boldsymbol{\theta}_t^{(i)} - \alpha(\boldsymbol{G}(\boldsymbol{\theta}_t^{(i)}) + \lambda\boldsymbol{I})^{-1}\nabla\mathcal{L}(\boldsymbol{\theta}_t^{(i)}), \qquad i = 1, \ldots, L. \tag{7}$$

We run the update rule with the following approximations of the generalized Gauss-Newton: the exact diagonal (DiagGGN) and an MC estimate (DiagGGN-MC), and the Kronecker factorizations KFAC (Martens & Grosse, 2015), KFLR and KFRA[5](Botev et al., 2017). The inversion required by the update rule is straightforward for the diagonal curvature. For the Kronecker-factored quantities, we use the approximation introduced by Martens & Grosse (2015) (see Appendix C.3).

These curvature estimates are tested for the training of deep neural networks by running the corresponding optimizers on the main test problems of the benchmarking suite DEEPOBS (Schneider et al., 2019).[6] We use the setup (batch size, number of training epochs) of DEEPOBS' baselines, and tune the learning rate $\alpha$ and damping parameter $\lambda$ with a grid search for each optimizer (details in Appendix C.2). The best hyperparameter settings is chosen according to the final accuracy on a validation set. We report the median and quartiles of the performance for ten random seeds.

Fig. 7a shows the results for the 3C3D network trained on CIFAR-10. The optimizers that leverage Kronecker-factored curvature approximations beat the baseline performance in terms of per-iteration progress on the training loss, training and test accuracy. Using the same hyperparameters, there is little difference between KFAC and KFLR, or DiagGGN and DiagGGN-MC. Given that the quantities based on MC-sampling are considerably cheaper, this experiment suggests it being an important technique for reducing the computational burden of curvature approximations.

Fig. 7b shows benchmarks for the ALL-CNN-C network trained on CIFAR-100. Due to the high-dimensional output, the curvatures using a full matrix propagation rather than an MC sample cannot be run on this problem due to memory issues. Both DiagGGN-MC and KFAC can compete with the baselines in terms of progress per iteration. As the update rule we implemented is simplistic on purpose, this is promising for future applications of second-order methods that can more efficiently use the additional information given by curvature approximations.

---

[5] KFRA was not originally designed for convolutions; we extend it using the Kronecker factorization of Grosse & Martens (2016). While it can be computed for small networks on MNIST, which we report in Appendix C.4, the approximate backward pass of KFRA does not seem to scale to large convolution layers.

[6] https://deepobs.github.io/. We cannot run BACKPACK on all test problems in this benchmark due to the limitations outlined in Section 2. Despite this limitation, we still run on models spanning a representative range of image classification problems.

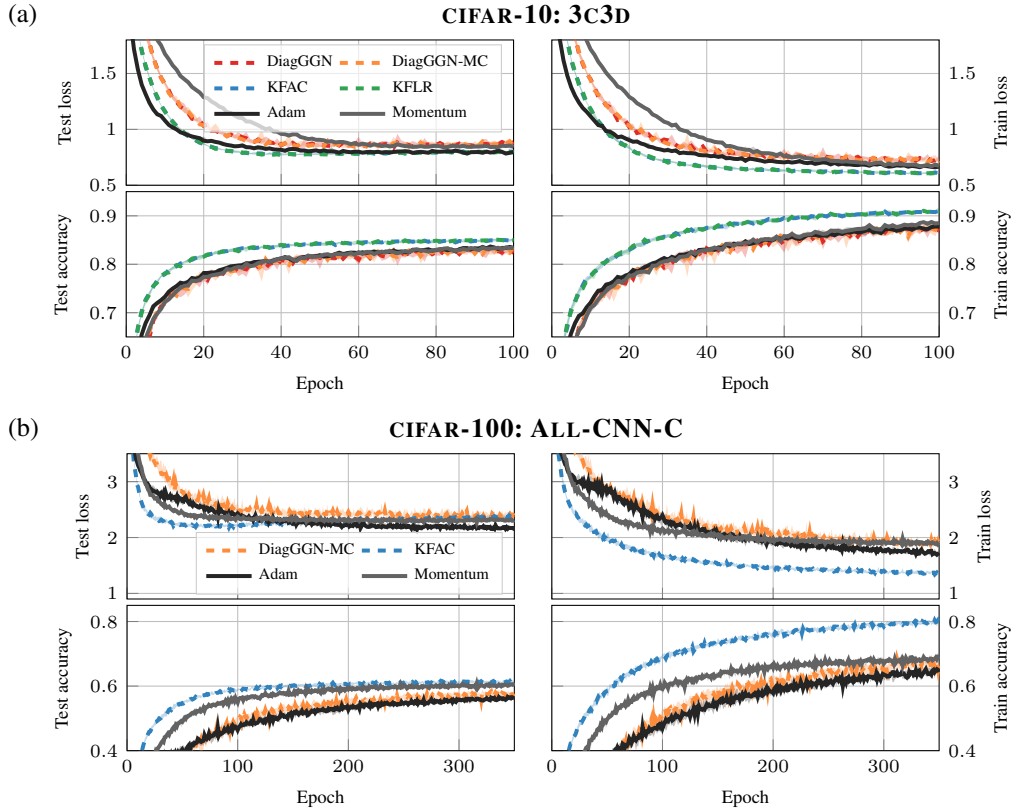

Figure 7: Median performance with shaded quartiles of the DEEPOBS benchmark for (a) 3C3D network (895,210 parameters) on CIFAR-10 and (b) ALL-CNN-C network (1,387,108 parameters) on CIFAR-100. Solid lines show baselines of momentum SGD and Adam provided by DEEPOBS.

## 5 CONCLUSION

Machine learning's coming-of-age has been accompanied, and in part driven, by a maturing of the software ecosystem. This has drastically simplified the lives of developers and researchers alike, but has also crystallized parts of the algorithmic landscape. This has dampened research in cutting-edge areas that are far from mature, like second-order optimization for deep neural networks. To ensure that good ideas can bear fruit, researchers must be able to compute new quantities without an overwhelming software development burden. To support research and development in optimization for deep learning, we have introduced BACKPACK, an efficient implementation in PYTORCH of recent conceptual advances and extensions to backpropagation (Tab. 1 lists all features). BACKPACK enriches the syntax of automatic differentiation packages to offer additional observables to optimizers beyond the batch-averaged gradient. Our experiments demonstrate that BACKPACK's implementation offers drastic efficiency gains over the kind of naïve implementation within reach of the typical researcher. As a demonstrative example, we "invented" a few optimization routines that, without BACKPACK, would require demanding implementation work and can now be tested with ease. We hope that studies like this allow BACKPACK to help mature the ML software ecosystem further.

## ACKNOWLEDGMENTS

The authors would like to thank Aaron Bahde, Ludwig Bald, and Frank Schneider for their help with DEEPOBS and Lukas Balles, Simon Bartels, Filip de Roos, Tim Fischer, Nicolas Krämer, Agustinus Kristiadi, Frank Schneider, Jonathan Wenger, and Matthias Werner for constructive feedback.

The authors gratefully acknowledge financial support by the European Research Council through ERC StG Action 757275 / PANAMA; the DFG Cluster of Excellence "Machine Learning - New

Table 1: Overview of the features supported in the first release of BACKPACK.

| Feature | Details |
|---|---|
| Individual gradients | $\frac{1}{N}\nabla_{\boldsymbol{\theta}^{(i)}}\ell_n(\boldsymbol{\theta}), \quad n=1,\ldots,N$ |
| Batch variance | $\frac{1}{N}\sum_{n=1}^{N}\left[\nabla_{\boldsymbol{\theta}^{(i)}}\ell_n(\boldsymbol{\theta})\right]_j^2 - \left[\nabla_{\boldsymbol{\theta}^{(i)}}\mathcal{L}(\boldsymbol{\theta})\right]_j^2$ |
| 2$^{\text{nd}}$ moment | $\frac{1}{N}\sum_{n=1}^{N}\left[\nabla_{\boldsymbol{\theta}^{(i)}}\ell_n(\boldsymbol{\theta})\right]_j^2, \quad j=1,\ldots,d^{(i)}.$ |
| Indiv. gradient $\ell_2$ norm | $\left\|\frac{1}{N}\nabla_{\boldsymbol{\theta}^{(i)}}\ell_n(\boldsymbol{\theta})\right\|_2^2, \quad n=1,\ldots,N$ |
| DiagGGN | $\mathrm{diag}\left(\boldsymbol{G}(\boldsymbol{\theta}^{(i)})\right)$ |
| DiagGGN-MC | $\mathrm{diag}\left(\tilde{\boldsymbol{G}}(\boldsymbol{\theta}^{(i)})\right)$ |
| Hessian diagonal | $\mathrm{diag}\left(\nabla_{\boldsymbol{\theta}^{(i)}}^2\mathcal{L}(\boldsymbol{\theta})\right)$ |
| KFAC | $\tilde{\boldsymbol{G}}(\boldsymbol{\theta}^{(i)}) \approx \boldsymbol{A}^{(i)} \otimes \boldsymbol{B}_{\text{KFAC}}^{(i)}$ |
| KFLR | $\boldsymbol{G}(\boldsymbol{\theta}^{(i)}) \approx \boldsymbol{A}^{(i)} \otimes \boldsymbol{B}_{\text{KFLR}}^{(i)}$ |
| KFRA | $\boldsymbol{G}(\boldsymbol{\theta}^{(i)}) \approx \boldsymbol{A}^{(i)} \otimes \boldsymbol{B}_{\text{KFRA}}^{(i)}$ |

Perspectives for Science", EXC 2064/1, project number 390727645; the German Federal Ministry of Education and Research (BMBF) through the Tübingen AI Center (FKZ: 01IS18039A); and funds from the Ministry of Science, Research and Arts of the State of Baden-Württemberg. F. D. is grateful to the International Max Planck Research School for Intelligent Systems (IMPRS-IS) for support.

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

# BACKPACK: PACKING MORE INTO BACKPROP
## SUPPLEMENTARY MATERIAL

**Table of Content**

## A   BACKPACK EXTENSIONS

This section provides more technical details on the additional quantities extracted by BACKPACK.

**Notation:** Consider an arbitrary module $T_{\boldsymbol{\theta}^{(i)}}^{(i)}$ of a network $i = 1, \dots, L$, parameterized by $\boldsymbol{\theta}^{(i)}$. It transforms the output of its parent layer for sample $n$, $\boldsymbol{z}_n^{(i-1)}$, to its output $\boldsymbol{z}_n^{(i)}$, i.e.

$$\boldsymbol{z}_n^{(i)} = T_{\boldsymbol{\theta}^{(i)}}^{(i)}(\boldsymbol{z}_n^{(i-1)}), \qquad n = 1, \dots, N, \tag{8}$$

where $N$ is the number of samples. In particular, $\boldsymbol{z}_n^{(0)} = \boldsymbol{x}_n$ and $\boldsymbol{z}_n^{(L)}(\boldsymbol{\theta}) = f(\boldsymbol{x}_n, \boldsymbol{\theta})$, where $f$ is the transformation of the whole network. The dimension of the hidden layer $i$'s output $\boldsymbol{z}_n^{(i)}$ is written $h^{(i)}$ and $\boldsymbol{\theta}^{(i)}$ is of dimension $d^{(i)}$. The dimension of the network output, the prediction $\boldsymbol{z}^{(L)}$, is $h^{(L)} = C$. For an image classification task, $C$ corresponds to the number of classes.

All quantities are assumed to be vector-shaped. For image-processing transformations that usually act on tensor-shaped inputs, we can reduce to the vector scenario by vectorizing all quantities; this discussion does not rely on a specific flattening scheme. However, for an efficient implementation, vectorization should match the layout of the memory of the underlying arrays.

**Jacobian:**   The Jacobian matrix $\mathrm{J}_{\boldsymbol{a}}\boldsymbol{b}$ of an arbitrary vector $\boldsymbol{b} \in \mathbb{R}^B$ with respect to another vector $\boldsymbol{a} \in \mathbb{R}^A$ is an $[A \times B]$ matrix of partial derivatives, $[\mathrm{J}_{\boldsymbol{a}}\boldsymbol{b}]_{ij} = \partial [\boldsymbol{b}]_i / \partial [\boldsymbol{a}]_j$.

### A.1   FIRST-ORDER QUANTITIES

The basis for the extraction of additional information about first-order derivatives is given by Eq. 3, which we state again for multiple samples,

$$\nabla_{\boldsymbol{\theta}^{(i)}}\mathcal{L}(\boldsymbol{\theta}) = \frac{1}{N}\sum_{n=1}^{N}\nabla_{\boldsymbol{\theta}^{(i)}}\ell_n(\boldsymbol{\theta}) = \frac{1}{N}\sum_{n=1}^{N}(\mathrm{J}_{\boldsymbol{\theta}^{(i)}}\boldsymbol{z}_n^{(i)})^{\top}(\nabla_{\boldsymbol{z}_n^{(i)}}\ell_n(\boldsymbol{\theta})).$$

During the backpropagation step of module $i$, we have access to $\nabla_{\boldsymbol{z}_n^{(i)}}\ell(\boldsymbol{\theta})$, $i = 1, \dots, N$. To extract more quantities involving the gradient, we use additional information about the transformation $T_{\boldsymbol{\theta}^{(i)}}^{(i)}$ within our custom implementation of the Jacobian $\mathrm{J}_{\boldsymbol{\theta}^{(i)}}\boldsymbol{z}_n^{(i)}$ and transposed Jacobian $(\mathrm{J}_{\boldsymbol{\theta}^{(i)}}\boldsymbol{z}_n^{(i)})^{\top}$.

**Individual gradients:**   The contribution of each sample to the overall gradient, $\frac{1}{N}\nabla_{\boldsymbol{\theta}^{(i)}}\ell_n(\boldsymbol{\theta})$, is computed by application of the transposed Jacobian,

$$\frac{1}{N}\nabla_{\boldsymbol{\theta}^{(i)}}\ell_n(\boldsymbol{\theta}) = \frac{1}{N}(\mathrm{J}_{\boldsymbol{\theta}^{(i)}}\boldsymbol{z}_n^{(i)})^{\top}(\nabla_{\boldsymbol{z}_n^{(i)}}\ell_n(\boldsymbol{\theta})), \qquad n = 1, \dots, N. \tag{9}$$

For each parameter $\boldsymbol{\theta}^{(i)}$ the individual gradients are of size $[N \times d^{(i)}]$.

**Individual gradient $\ell_2$ norm:**   The quantity $\left\|\frac{1}{N}\nabla_{\boldsymbol{\theta}^{(i)}}\ell_n(\boldsymbol{\theta})\right\|_2^2$, for $n = 1, ..., N$, could be extracted from the individual gradients (Eq. 9) as

$$\left\|\frac{1}{N}\nabla_{\boldsymbol{\theta}^{(i)}}\ell_n(\boldsymbol{\theta})\right\|_2^2 = \left[\frac{1}{N}(\mathrm{J}_{\boldsymbol{\theta}^{(i)}}\boldsymbol{z}_n^{(i)})^\top(\nabla_{\boldsymbol{z}_n^{(i)}}\ell_n(\boldsymbol{\theta}))\right]^\top \left[\frac{1}{N}(\mathrm{J}_{\boldsymbol{\theta}^{(i)}}\boldsymbol{z}_n^{(i)})^\top(\nabla_{\boldsymbol{z}_n^{(i)}}\ell_n(\boldsymbol{\theta}))\right],$$

which is an $N$-dimensional object for each parameter $\boldsymbol{\theta}^{(i)}$. However, this is not memory efficient as the individual gradients are an $[N \times d^{(i)}]$ tensor. To circumvent this problem, BACKPACK uses the structure of the Jacobian whenever possible.

For a specific example, take a linear layer with parameters $\boldsymbol{\theta}$ as an $[A \times B]$ matrix. The layer transforms the inputs $\boldsymbol{z}_n^{(i-1)}$, an $[N \times A]$ matrix which we will now refer to as $\boldsymbol{A}$. During the backward pass, it receives the gradient of the individual losses with respect to its output, $\{\frac{1}{N}\nabla_{\boldsymbol{z}_n^{(i)}}\ell_n\}_{n=1}^N$, as an $[N \times B]$ matrix which we will refer to as $\boldsymbol{B}$. The overall gradient, an $[A \times B]$ matrix, can be computed as $\boldsymbol{A}^\top\boldsymbol{B}$, and the individual gradients are a set of $N$ $[A \times B]$ matrices, $\{\boldsymbol{A}[n,:]\boldsymbol{B}[n,:]^\top\}_{n=1}^N$. We want to avoid storing that information. To reduce the memory requirement, note that the individual gradient norm can be written as

$$\left\|\frac{1}{N}\nabla_{\boldsymbol{\theta}}\ell_n\right\|^2 = \sum_i\sum_j(\boldsymbol{A}[n,i]\boldsymbol{B}[n,j])^2,$$

and that the summation can be done independently for each matrix, as $\sum_i\sum_j(\boldsymbol{A}[n,i]\boldsymbol{B}[n,j])^2 = (\sum_i\boldsymbol{A}[n,i])^2(\sum_j\boldsymbol{B}[n,j]^2)$. Therefore, we can square each matrix (element-wise) and sum over non-batch dimensions. This yields vectors $\boldsymbol{a}, \boldsymbol{b}$ of $N$ elements, where $\boldsymbol{a}[n] = \sum_i\boldsymbol{A}[n,i]^2$. The individual gradients' $\ell_2$ norm is then given by $\boldsymbol{a}\circ\boldsymbol{b}$ where $\circ$ is element-wise multiplication.

**Second moment:**   The gradient second moment (or more specifically, the diagonal of the second moment) is the sum of the squared elements of the individual gradients in a mini-batch, i.e.

$$\frac{1}{N}\sum_{n=1}^N[\nabla_{\boldsymbol{\theta}^{(i)}}\ell_n(\boldsymbol{\theta})]_j^2, \qquad j = 1, \ldots, d^{(i)}. \tag{10}$$

It can be used to evaluate the variance of individual elements of the gradient (see below). The second moment is of dimension $d^{(i)}$, the same dimension as the layer parameter $\boldsymbol{\theta}^{(i)}$. Similarly to the $\ell_2$ norm, it can be computed from individual gradients, but is more efficiently computed implicitly.

Revisiting the example of the linear layer from the individual $\ell_2$ norm computation, the second moment of the parameters $\boldsymbol{\theta}[i,j]$ is given by $\sum_n(\boldsymbol{A}[n,i]\boldsymbol{B}[n,j])^2$, which can be directly computed by taking the element-wise square of $\boldsymbol{A}$ and $\boldsymbol{B}$ element-wise, $\boldsymbol{A}^2, \boldsymbol{B}^2$, and computing $\boldsymbol{A}^{2\top}\boldsymbol{B}^2$.

**Variance:**   Gradient variances over a mini-batch (or more precisely, the diagonal of the covariance) can be computed using the second moment and the gradient itself,

$$\frac{1}{N}\sum_{n=1}^N[\nabla_{\boldsymbol{\theta}^{(i)}}\ell_n(\boldsymbol{\theta})]_j^2 - [\nabla_{\boldsymbol{\theta}^{(i)}}\mathcal{L}(\boldsymbol{\theta})]_j^2, \qquad j = 1, \ldots, d^{(i)}. \tag{11}$$

The element-wise gradient variance of same dimension as the layer parameter $\boldsymbol{\theta}^{(i)}$, i.e. $d^{(i)}$.

## A.2   Second-order quantities based on the generalized Gauss-Newton

The computation of quantities that originate from the approximations of the Hessian require an additional backward pass (see Dangel et al. (2019)). Most curvature approximations supported by BACKPACK rely on the generalized Gauss-Newton (GGN) matrix (Schraudolph, 2002)

$$\boldsymbol{G}(\boldsymbol{\theta}) = \frac{1}{N}\sum_{n=1}^N(\mathrm{J}_{\boldsymbol{\theta}}f(\boldsymbol{x}_n, \boldsymbol{\theta}))^\top\nabla_f^2\ell(f(\boldsymbol{x}_n, \boldsymbol{\theta}), \boldsymbol{y}_n)(\mathrm{J}_{\boldsymbol{\theta}}f(\boldsymbol{x}_n, \boldsymbol{\theta})). \tag{12}$$

One interpretation of the GGN is that it corresponds to the empirical risk Hessian when the model $f$ is approximated with its first-order Taylor expansion, i.e. by linearizing the network and ignoring

second-order effects. Hence, the effect of module curvature in the recursive scheme of Eq. 4 can be ignored to obtain the simpler expression

$$
\begin{aligned}
\boldsymbol{G}(\boldsymbol{\theta}^{(i)}) &= \frac{1}{N} \sum_{n=1}^{N} (\mathrm{J}_{\boldsymbol{\theta}^{(i)}} f)^{\top} \nabla_f^2 \ell(f(\boldsymbol{x}_n, \boldsymbol{\theta}), \boldsymbol{y}_n)(\mathrm{J}_{\boldsymbol{\theta}^{(i)}} f) \\
&= \frac{1}{N} \sum_{n=1}^{N} (\mathrm{J}_{\boldsymbol{\theta}^{(i)}} \boldsymbol{z}_n^{(i)})^{\top} \boldsymbol{G}(\boldsymbol{z}_n^{(i)})(\mathrm{J}_{\boldsymbol{\theta}^{(i)}} \boldsymbol{z}_n^{(i)})
\end{aligned}
\tag{13}
$$

for the exact block diagonal of the full GGN. In analogy to $\boldsymbol{G}(\boldsymbol{\theta}^{(i)})$ we have introduced the $[d^{(i)} \times d^{(i)}]$-dimensional quantity

$$
\boldsymbol{G}(\boldsymbol{z}_n^{(i)}) = (\mathrm{J}_{\boldsymbol{z}_n^{(i)}} f)^{\top} \nabla_f^2 \ell(f(\boldsymbol{x}_n, \boldsymbol{\theta}), \boldsymbol{y}_n)(\mathrm{J}_{\boldsymbol{z}_n^{(i)}} f)
$$

that needs to be backpropagated. The curvature backpropagation also follows from Eq. 4 as

$$
\boldsymbol{G}(\boldsymbol{z}_n^{(i-1)}) = (\mathrm{J}_{\boldsymbol{z}_n^{(i-1)}} \boldsymbol{z}_n^{(i)})^{\top} \boldsymbol{G}(\boldsymbol{z}_n^{(i)})(\mathrm{J}_{\boldsymbol{z}_n^{(i-1)}} \boldsymbol{z}_n^{(i)}), \qquad i = 1, \dots, L,
\tag{14a}
$$

and is initialized with the Hessian of the loss function with respect to the network prediction, i.e.

$$
\boldsymbol{G}(\boldsymbol{z}_n^{(L)}) = \nabla_f^2 \ell(f(\boldsymbol{x}_n, \boldsymbol{\theta}), \boldsymbol{y}_n).
\tag{14b}
$$

Although this scheme is exact, it is computationally infeasible as it requires the backpropagation of $N$ $[h^{(i)} \times h^{(i)}]$ matrices between module $i + 1$ and $i$. Even for small $N$, this is not possible for networks containing large convolutions.

As an example, the first layer of the ALL-CNN-C network outputs $29 \times 29$ images with 96 channels, which already gives $h^{(i)} = 80{,}736$, which leads to half a Gigabyte per sample. Moreover, storing all the $[d^{(i)} \times d^{(i)}]$-dimensional blocks $\boldsymbol{G}(\boldsymbol{\theta}^{(i)})$ is not possible. BACKPACK implements different approximation strategies, developed by Martens & Grosse (2015) and Botev et al. (2017) that address both of these complexity issues from different perspectives.

**Symmetric factorization scheme:** One way to improve the memory footprint of the backpropagated matrices in the case where the model prediction's dimension $C$ (the number of classes in an image classification task) is small compared to all hidden features $h^{(i)}$ is to propagate a symmetric factorization of the GGN instead. It relies on the observation that if the loss function itself is convex, even though its composition with the network might not be, its Hessian with respect to the network output can be decomposed as

$$
\nabla_f^2 \ell(f(\boldsymbol{x}_n, \boldsymbol{\theta}), \boldsymbol{y}_n) = \boldsymbol{S}(\boldsymbol{z}_n^{(L)}) \boldsymbol{S}(\boldsymbol{z}_n^{(L)})^{\top}
\tag{15}
$$

with the $[C \times C]$-dimensional matrix factorization of the loss Hessian, $\boldsymbol{S}(\boldsymbol{z}_n^{(L)})$, for sample $n$. Consequently, the GGN in Eq. 12 reduces to an outer product,

$$
\boldsymbol{G}(\boldsymbol{\theta}) = \frac{1}{N} \sum_{n=1}^{N} \left[ (\mathrm{J}_{\boldsymbol{\theta}} f)^{\top} \boldsymbol{S}(\boldsymbol{z}_n^{(L)}) \right] \left[ (\mathrm{J}_{\boldsymbol{\theta}} f)^{\top} \boldsymbol{S}(\boldsymbol{z}_n^{(L)}) \right]^{\top}.
\tag{16}
$$

The analogue for diagonal blocks follows from Eq. 13 and reads

$$
\boldsymbol{G}(\boldsymbol{\theta}^{(i)}) = \frac{1}{N} \sum_{n=1}^{N} \left[ (\mathrm{J}_{\boldsymbol{\theta}^{(i)}} \boldsymbol{z}_n^{(i)})^{\top} \boldsymbol{S}(\boldsymbol{z}_n^{(i)}) \right] \left[ (\mathrm{J}_{\boldsymbol{\theta}^{(i)}} \boldsymbol{z}_n^{(i)})^{\top} \boldsymbol{S}(\boldsymbol{z}_n^{(i)}) \right]^{\top},
\tag{17}
$$

where we defined the $[h^{(i)} \times C]$-dimensional matrix square root $\boldsymbol{S}(\boldsymbol{z}_n^{(i)}) = (\mathrm{J}_{\boldsymbol{z}_n^{(i)}} f)^{\top} \boldsymbol{S}(\boldsymbol{z}_n^{(L)})$. Instead of having layer $i$ backpropagate $N$ objects of shape $[h^{(i)} \times h^{(i)}]$ according to Eq. 14, we instead backpropagate the matrix square root via

$$
\boldsymbol{S}(\boldsymbol{z}_n^{(i-1)}) = (\mathrm{J}_{\boldsymbol{z}_n^{(i-1)}} \boldsymbol{z}_n^{(i)})^{\top} \boldsymbol{S}(\boldsymbol{z}_n^{(i)})(\mathrm{J}_{\boldsymbol{z}_n^{(i-1)}} \boldsymbol{z}_n^{(i)}), \qquad i = 1, \dots, L,
\tag{18}
$$

starting with Eq. 15. This reduces the backpropagated matrix of layer $i$ to $[h^{(i)} \times C]$ for each sample.

A.2.1 DIAGONAL CURVATURE APPROXIMATIONS

**Diagonal of the GGN (DiagGGN):**  The factorization trick for the loss Hessian reduces the size of the backpropagated quantities, but does not address the intractable size of the GGN diagonal blocks $\boldsymbol{G}(\boldsymbol{\theta}^{(i)})$. In BACKPACK, we can extract $\mathrm{diag}\left(\boldsymbol{G}(\boldsymbol{\theta}^{(i)})\right)$ given the backpropagated quantities $\boldsymbol{S}(\boldsymbol{z}_n^{(i)})$, $i = 1, \ldots, N$, without building up the matrix representation of Eq. 17. In particular, we compute

$$\mathrm{diag}\left(\boldsymbol{G}(\boldsymbol{\theta}^{(i)})\right) = \frac{1}{N} \sum_{n=1}^{N} \mathrm{diag}\left(\left[(\mathrm{J}_{\boldsymbol{\theta}^{(i)}} \boldsymbol{z}_n^{(i)})^\top \boldsymbol{S}(\boldsymbol{z}_n^{(i)})\right] \left[(\mathrm{J}_{\boldsymbol{\theta}^{(i)}} \boldsymbol{z}_n^{(i)})^\top \boldsymbol{S}(\boldsymbol{z}_n^{(i)})\right]^\top \right) . \tag{19}$$

**Diagonal of the GGN with MC sampled loss Hessian (DiagGGN-MC):**  We use the same backpropagation strategy of Eq. 18, replacing the symmetric factorization of Eq. 15 with an approximation by a smaller matrix $\tilde{\boldsymbol{S}}(\boldsymbol{z}_n^{(L)})$ of size $[C \times \tilde{C}]$ and $\tilde{C} < C$,

$$\nabla_f^2 \ell(f(\boldsymbol{x}_n, \boldsymbol{\theta}), \boldsymbol{y}_n) \approx \tilde{\boldsymbol{S}}(\boldsymbol{z}_n^{(L)}) \left(\tilde{\boldsymbol{S}}(\boldsymbol{z}_n^{(L)})\right)^\top . \tag{20}$$

This further reduces the size of backpropagated curvature quantities. Martens & Grosse (2015) introduced such a sampling scheme with KFAC based on the connection between the GGN and the Fisher. Most loss functions used in machine learning have a probabilistic interpretation as negative log-likelihood of a probabilistic model. The squarred error of regression is equivalent to a Gaussian noise assumption and the cross-entropy is linked to the categorical distribution. In this case, the loss Hessian with respect to the network output is equal, in expectation, to the outer products of gradients *if the output of the network is sampled according to a particular distribution*, $p_f(\boldsymbol{x})$, defined by the network output $f(\boldsymbol{x})$. Sampling outputs $\hat{\boldsymbol{y}} \sim p$, we have that

$$\mathbb{E}_{\hat{y} \sim p_{f(\boldsymbol{x})}} \left[\nabla_{\boldsymbol{\theta}} \ell(f(\boldsymbol{x}, \boldsymbol{\theta}), \hat{\boldsymbol{y}}) \nabla_{\boldsymbol{\theta}} \ell(f(\boldsymbol{x}, \boldsymbol{\theta}), \hat{\boldsymbol{y}})^\top\right] = \nabla_{\boldsymbol{\theta}}^2 \ell(f(\boldsymbol{x}, \boldsymbol{\theta}), \boldsymbol{y}) . \tag{21}$$

Sampling one such gradient leads to a rank-1 MC approximation of the loss Hessian. With the substitution $\boldsymbol{S} \leftrightarrow \tilde{\boldsymbol{S}}$, we compute an MC approximation of the GGN diagonal in BACKPACK as

$$\mathrm{diag}\left(\boldsymbol{G}(\boldsymbol{\theta}^{(i)})\right) \approx \frac{1}{N} \sum_{n=1}^{N} \mathrm{diag}\left(\left[(\mathrm{J}_{\boldsymbol{\theta}^{(i)}} \boldsymbol{z}_n^{(i)})^\top \tilde{\boldsymbol{S}}(\boldsymbol{z}_n^{(i)})\right] \left[(\mathrm{J}_{\boldsymbol{\theta}^{(i)}} \boldsymbol{z}_n^{(i)})^\top \tilde{\boldsymbol{S}}(\boldsymbol{z}_n^{(i)})\right]^\top \right) . \tag{22}$$

A.2.2 KRONECKER-FACTORED CURVATURE APPROXIMATIONS

A different approach to reduce memory complexity of the GGN blocks $\boldsymbol{G}(\boldsymbol{\theta}^{(i)})$, apart from diagonal curvature approximations, is representing them as Kronecker products (KFAC for linear and convolution layers by Martens & Grosse (2015); Grosse & Martens (2016) KFLR and KFRA for linear layers by Botev et al. (2017)),

$$\boldsymbol{G}(\boldsymbol{\theta}^{(i)}) = \boldsymbol{A}^{(i)} \otimes \boldsymbol{B}^{(i)} . \tag{23}$$

For both linear and convolution layers, the first Kronecker factor $\boldsymbol{A}^{(i)}$ is obtained from the inputs $\boldsymbol{z}_n^{(i-1)}$ to layer $i$. Instead of repeating the technical details of the aforementioned references, we will focus on how they differ in (i) the backpropagated quantities and (ii) the backpropagation strategy. As a result, we will be able to extend KFLR and KFRA to convolutional neural networks[7].

**KFAC and KFLR:**  KFAC uses an MC-sampled estimate of the loss Hessian with a square root factorization $\tilde{\boldsymbol{S}}(\boldsymbol{z}_n^{(L)})$ like in Eq. 20. The backpropagation is equivalent to the computation of the GGN diagonal. For the GGN of the weights of a linear layer $i$, the second Kronecker term is given by

$$\boldsymbol{B}_{\mathrm{KFAC}}^{(i)} = \frac{1}{N} \sum_{n=1}^{N} \tilde{\boldsymbol{S}}(\boldsymbol{z}_n^{(i)}) \left(\tilde{\boldsymbol{S}}(\boldsymbol{z}_n^{(i)})\right)^\top ,$$

---

[7]We keep the PYTORCH convention that weights and bias are treated as separate parameters. For the bias terms, we can store the full matrix representation of the GGN. This factor reappears in the Kronecker factorization of the GGN with respect to the weights.

which at the same time corresponds to the GGN of the layer's bias[8].

In contrast to KFAC, the KFLR approximation backpropagates the exact square root factorization $\boldsymbol{S}(\boldsymbol{z}_n^{(L)})$, i.e. for the weights of a linear layer[8] (see Botev et al. (2017) for more details)

$$\boldsymbol{B}_{\text{KFLR}}^{(i)} = \frac{1}{N} \sum_{n=1}^{N} \boldsymbol{S}(\boldsymbol{z}_n^{(i)}) \left( \boldsymbol{S}(\boldsymbol{z}_n^{(i)}) \right)^{\top} .$$

**KFRA:** The backpropagation strategy for KFRA eliminates the scaling of the backpropagated curvature quantities with the batch size $N$ in Eq. 14. Instead of having layer $i$ receive the $N$ exact $[h^{(i)} \times h^{(i)}]$ matrices $\boldsymbol{G}(\boldsymbol{z}_n^{(i)})$, $n = 1, \ldots, N$, only a single averaged object, denoted $\overline{\boldsymbol{G}}^{(i)}$, is used as an approximation. In particular, the recursion changes to

$$\overline{\boldsymbol{G}}^{(i-1)} = \frac{1}{N} \sum_{n=1}^{N} (\text{J}_{\boldsymbol{z}_n^{(i-1)}} \boldsymbol{z}_n^{(i)})^{\top} \overline{\boldsymbol{G}}^{(i)} (\text{J}_{\boldsymbol{z}_n^{(i-1)}} \boldsymbol{z}_n^{(i)}) , \qquad i = 1, \ldots, L , \tag{24a}$$

and is initialized with the batch-averaged loss Hessian

$$\overline{\boldsymbol{G}}^{(L)} = \frac{1}{N} \sum_{n=1}^{N} \nabla_f^2 \ell(f(\boldsymbol{x}_n, \boldsymbol{\theta}), \boldsymbol{y}_n) . \tag{24b}$$

For a linear layer, KFRA uses[8] (see Botev et al. (2017) for more details)

$$\boldsymbol{B}_{\text{KFRA}}^{(i)} = \overline{\boldsymbol{G}}^{(i)}.$$

## A.3 THE EXACT HESSIAN DIAGONAL

For neural networks consisting only of piecewise linear activation functions, computing the diagonal of the Hessian is equivalent to computing the GGN diagonal. This is because for these activations the second term in the Hessian backpropagation recursion (Eq. 4) vanishes.

However, for activation functions with non-vanishing second derivative, these residual terms have to be accounted for in the backpropagation. The Hessian backpropagation for module $i$ reads

$$\nabla_{\boldsymbol{\theta}^{(i)}}^2 \ell(\boldsymbol{\theta}) = (\text{J}_{\boldsymbol{\theta}^{(i)}} \boldsymbol{z}_n^{(i)})^{\top} (\nabla_{\boldsymbol{z}_n^{(i)}}^2 \ell(\boldsymbol{\theta})) (\text{J}_{\boldsymbol{\theta}^{(i)}} \boldsymbol{z}_n^{(i)}) + \boldsymbol{R}_n^{(i)}(\boldsymbol{\theta}^{(i)}) , \tag{25a}$$

$$\nabla_{\boldsymbol{z}_n^{(i-1)}}^2 \ell(\boldsymbol{\theta}) = (\text{J}_{\boldsymbol{z}_n^{(i-1)}} \boldsymbol{z}_n^{(i)})^{\top} (\nabla_{\boldsymbol{z}_n^{(i)}}^2 \ell(\boldsymbol{\theta})) (\text{J}_{\boldsymbol{z}_n^{(i-1)}} \boldsymbol{z}_n^{(i)}) + \boldsymbol{R}_n^{(i)}(\boldsymbol{z}_n^{(i-1)}) , \tag{25b}$$

for $n = 1, \ldots, N$. Those $[h^{(i)} \times h^{(i)}]$-dimensional residual terms are defined as

$$\boldsymbol{R}_n^{(i)}(\boldsymbol{\theta}^{(i)}) = \sum_j \left( \nabla_{\boldsymbol{\theta}^{(i)}}^2 [\boldsymbol{z}_n^{(i)}]_j \right) \left[ \nabla_{\boldsymbol{z}_n^{(i)}} \ell(\boldsymbol{\theta}) \right]_j ,$$

$$\boldsymbol{R}_n^{(i)}(\boldsymbol{z}_n^{(i-1)}) = \sum_j \left( \nabla_{\boldsymbol{z}_n^{(i-1)}}^2 [\boldsymbol{z}_n^{(i)}]_j \right) \left[ \nabla_{\boldsymbol{z}_n^{(i)}} \ell(\boldsymbol{\theta}) \right]_j ,$$

For common parameterized layers, such as linear and convolution transformations, $\boldsymbol{R}_n^{(i)}(\boldsymbol{\theta}^{(i)}) = 0$. If the activation function is applied element-wise, $\boldsymbol{R}_n^{(i)}(\boldsymbol{z}_n^{(i-1)})$ are diagonal matrices.

Storing these quantities becomes very memory-intensive for high-dimensional nonlinear activation layers. In BACKPACK, this complexity is reduced by application of the aforementioned matrix square root factorization trick. To do so, we express the symmetric factorization of $\boldsymbol{R}_n^{(i)}(\boldsymbol{z}_n^{(i-1)})$ as

$$\boldsymbol{R}_n^{(i)}(\boldsymbol{z}_n^{(i-1)}) = \boldsymbol{P}_n^{(i)}(\boldsymbol{z}_n^{(i-1)}) \left( \boldsymbol{P}_n^{(i)}(\boldsymbol{z}_n^{(i-1)}) \right)^{\top} - \boldsymbol{N}_n^{(i)}(\boldsymbol{z}_n^{(i-1)}) \left( \boldsymbol{N}_n^{(i)}(\boldsymbol{z}_n^{(i-1)}) \right)^{\top} , \tag{26}$$

where $\boldsymbol{P}_n^{(i)}(\boldsymbol{z}_n^{(i-1)})$, $\boldsymbol{N}_n^{(i)}(\boldsymbol{z}_n^{(i-1)})$ represent the matrix square root of $\boldsymbol{R}_n^{(i)}(\boldsymbol{z}_n^{(i-1)})$ projected on its positive and negative eigenspace, respectively.

---

[8]In the case of convolutions, one has to sum over the spatial indices of a single channel of $\boldsymbol{z}_n^{(i)}$ as the bias is added to an entire channel, see Grosse & Martens (2016) for details.

This composition allows for the extension of the GGN backpropagation: In addition to $\boldsymbol{S}(\boldsymbol{z}_n^{(i)})$, the decompositions $\boldsymbol{P}_n^{(i)}(\boldsymbol{z}_n^{(i-1)}), \boldsymbol{N}_n^{(i)}(\boldsymbol{z}_n^{(i-1)})$ for the residual parts also have to be backpropagated according to Eq. 18. All diagonals are extracted from the backpropagated matrix square roots (see Eq. 19). All diagonals stemming from decompositions in the negative residual eigenspace have to be weighted by a factor of $-1$ before summation.

In terms of complexity, one backpropagation for $\boldsymbol{R}_n^{(i)}(\boldsymbol{z}^{(i-1)})$ changes the dimensionality as follows

$$\boldsymbol{R}_n^{(i)}(\boldsymbol{z}^{(i-1)}): \qquad [h^{(i)} \times h^{(i)}] \to [h^{(i-1)} \times h^{(i-1)}] \to [h^{(i-2)} \times h^{(i-2)}] \to \dots .$$

With the square root factorization, one instead obtains

$$\boldsymbol{P}_n^{(i)}(\boldsymbol{z}_n^{(i-1)}): \qquad [h^{(i)} \times h^{(i)}] \to [h^{(i-1)} \times h^{(i)}] \to [h^{(i-2)} \times h^{(i)}] \to \dots ,$$
$$\boldsymbol{N}_n^{(i)}(\boldsymbol{z}_n^{(i-1)}): \qquad [h^{(i)} \times h^{(i)}] \to [h^{(i-1)} \times h^{(i)}] \to [h^{(i-2)} \times h^{(i)}] \to \dots .$$

Roughly speaking, this scheme is more efficient whenever the hidden dimension of a nonlinear activation layer deceeds the largest hidden dimension of the network.

**Example:** Consider one backpropagation step of module $i$. Assume $\boldsymbol{R}_n^{(i)}(\boldsymbol{\theta}^{(i)}) = 0$, i.e. a linear, convolution, or non-parameterized layer. Then the following computations are performed in the protocol for the diagonal Hessian:

- Receive the following quantities from the child module $i+1$ (for $n = 1, \dots, N$)

$$\Phi = \Big\{ \boldsymbol{S}(\boldsymbol{z}_n^{(i)}),$$
$$\boldsymbol{P}_n^{(i+1)}(\boldsymbol{z}_n^{(i)}),$$
$$\boldsymbol{N}_n^{(i+1)}(\boldsymbol{z}_n^{(i)}),$$
$$(\mathrm{J}_{\boldsymbol{z}_n^{(i)}}\boldsymbol{z}_n^{(i+1)})^\top \boldsymbol{P}_n^{(i+2)}(\boldsymbol{z}_n^{(i+1)}),$$
$$(\mathrm{J}_{\boldsymbol{z}_n^{(i)}}\boldsymbol{z}_n^{(i+1)})^\top \boldsymbol{N}_n^{(i+2)}(\boldsymbol{z}_n^{(i+1)}),$$
$$\dots$$
$$(\mathrm{J}_{\boldsymbol{z}_n^{(i)}}\boldsymbol{z}_n^{(i+1)})^\top (\mathrm{J}_{\boldsymbol{z}_n^{(i+1)}}\boldsymbol{z}_n^{(i+2)})^\top \dots (\mathrm{J}_{\boldsymbol{z}_n^{(L-3)}}\boldsymbol{z}_n^{(L-2)})^\top \boldsymbol{P}_n^{(L-1)}(\boldsymbol{z}_n^{(L-2)}),$$
$$(\mathrm{J}_{\boldsymbol{z}_n^{(i)}}\boldsymbol{z}_n^{(i+1)})^\top (\mathrm{J}_{\boldsymbol{z}_n^{(i+1)}}\boldsymbol{z}_n^{(i+2)})^\top \dots (\mathrm{J}_{\boldsymbol{z}_n^{(L-3)}}\boldsymbol{z}_n^{(L-2)})^\top \boldsymbol{N}_n^{(L-1)}(\boldsymbol{z}_n^{(L-2)}) \Big\}$$

- Extract the module parameter Hessian diagonal, $\mathrm{diag}\left(\nabla_{\boldsymbol{\theta}^{(i)}}^2 \mathcal{L}(\boldsymbol{\theta})\right)$
  - For each quantity $\boldsymbol{A} \in \Phi$ extract the diagonal from the square root factorization and sum over the samples, i.e. compute

$$\frac{1}{N} \sum_{n=1}^N \mathrm{diag}\left( \left[(\mathrm{J}_{\boldsymbol{\theta}^{(i)}}\boldsymbol{z}_n^{(i)})^\top \boldsymbol{A}_n\right] \left[(\mathrm{J}_{\boldsymbol{\theta}^{(i)}}\boldsymbol{z}_n^{(i)})^\top \boldsymbol{A}_n\right]^\top \right) .$$

  Multiply the expression by $-1$ if $\boldsymbol{A}$ stems from backpropagation of a residual's negative eigenspace's factorization.
  - Sum all expressions to obtain the block Hessian's diagonal $\mathrm{diag}\left(\nabla_{\boldsymbol{\theta}^{(i)}}^2 \mathcal{L}(\boldsymbol{\theta})\right)$
- Backpropagate the received quantities to the parent module $i-1$
  - For each quantity $\boldsymbol{A}_n \in \Phi$, apply $(\mathrm{J}_{\boldsymbol{z}_n^{(i-1)}}\boldsymbol{z}_n^{(i)})^\top \boldsymbol{A}_n$
  - Append $\boldsymbol{P}_n^{(i+1)}(\boldsymbol{z}_n^{(i)})$ and $\boldsymbol{N}_n^{(i+1)}(\boldsymbol{z}_n^{(i)})$ to $\Phi$

# B  ADDITIONAL DETAILS ON BENCHMARKS

**KFAC VS. KFLR:** As the KFLR of Botev et al. (2017) is orders of magnitude more expensive to compute than the KFAC of Martens & Grosse (2015) on CIFAR-100, it was not included in the

main plot. This is not an implementation error; it follows from the definition of those methods. To approximate the GGN, $G(\theta) = \sum_n [J_\theta f_n]^\top \nabla^2_{f_n} \ell_n [J_\theta f_n]$, KFAC uses a rank-1 approximation for each of the inner Hessian $\nabla^2_{f_n} \ell_n = s_n s_n^\top$, and needs to propagate a *vector* through the computation graph for each sample. KFLR uses the complete inner Hessian instead. For CIFAR-100, the network has 100 output nodes—one for each class—and the inner Hessians are $[100 \times 100]$ matrices. KFLR needs to propagate a *matrix* through the computation graph for each sample, which is $100\times$ more expensive as shown in Fig. 8.

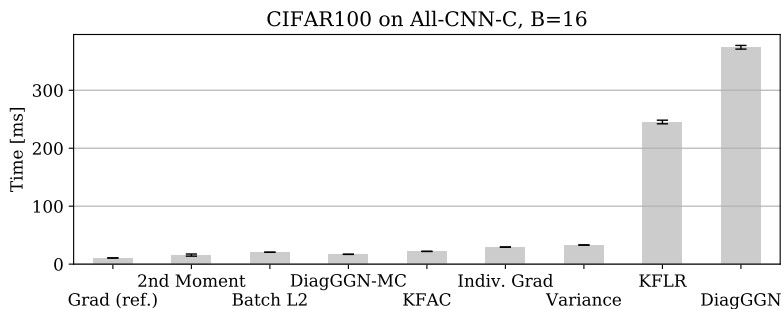

Figure 8: KFLR and DiagGGN are more expensive to run on large networks. The gradient takes less than $20\,\text{ms}$ to compute, but KFLR and DiagGGN are approximately $100\times$ more expensive.

**Diagonal of the GGN vs. Diagonal of the Hessian:** Most networks used in deep learning use ReLU activation functions. ReLU functions have no *curvature* as they are piecewise linear. Because of this, the diagonal of the GGN is equivalent to the diagonal of the Hessian (Martens, 2014). However, for networks that use non piecewise linear activation functions like sigmoids or tanh, computing the Hessian diagonal can be much more expensive than the GGN diagonal. To illustrate this point, we modify the smaller network used in our benchmarks to include a single sigmoid activation function before the last classification layer. The results in Fig. 9 show that the computation of the diagonal of the Hessian is already an order of magnitude more expensive than for the GGN.

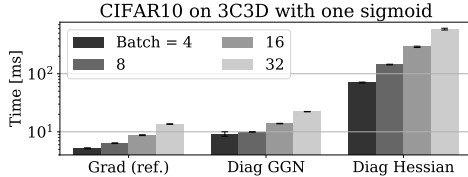

Figure 9: **Diagonal of the Hessian vs. the GGN.** If the network contains a single sigmoid activation function, the diagonal of the Hessian is an order of magnitude more computationally intensive than the diagonal of the GGN.

## C ADDITIONAL DETAILS ON EXPERIMENTS

### C.1 PROTOCOL

The optimizer experiments are performed according to the protocol suggested by DEEPOBS:

- Train the neural network with the investigated optimizer and vary its hyperparameters on a specified grid. This training is performed for a single random seed only.

- DEEPOBS evaluates metrics during the training procedure. From all runs of the grid search, it selects the best run automatically. The results shown in this work were obtained with the default strategy, favoring highest final accuracy on the validation set.

- For a better understanding of the optimizer performance with respect to randomized routines in the training process, DEEPOBS reruns the best hyperparameter setting for ten different random seeds. The results show mean values over these repeated runs, with standard deviations as uncertainty indicators.

- Along with the benchmarked optimizers, we show the DEEPOBS base line performances for Adam and momentum SGD (Momentum). They are provided by DEEPOBS.

The optimizers built upon BACKPACK's curvature estimates were benchmarked on the DEEPOBS image classification problems summarized in Table 2.

Table 2: Test problems considered from the DEEPOBS library (Schneider et al., 2019).

| Codename | Description | Dataset | # Parameters |
|---|---|---|---|
| LOGREG | Linear model | MNIST | 7,850 |
| 2C2D | 2 convolutional and 2 dense linear layers | FASHION-MNIST | 3,274,634 |
| 3C3D | 3 convolutional and 3 dense linear layers | CIFAR-10 | 895,210 |
| ALL-CNN-C | 9 convolutional layers (Springenberg et al., 2015) | CIFAR-100 | 1,387,108 |

## C.2 GRID SEARCH AND BEST HYPERPARAMETER SETTING

Both the learning rate $\alpha$ and damping $\lambda$ are tuned over the grid

$$\alpha \in \left\{10^{-4}, 10^{-3}, 10^{-2}, 10^{-1}, 1\right\}, \quad \lambda \in \left\{10^{-4}, 10^{-3}, 10^{-2}, 10^{-1}, 1, 10\right\}.$$

We use the same batch size ($N = 128$ for all problems, except $N = 256$ for ALL-CNN-C on CIFAR-100) as the base lines and the optimizers run for the identical number of epochs.

The best hyperparameter settings are summarized in Table 3.

## C.3 UPDATE RULE

We use a simple update rule with a constant damping parameter $\lambda$. Consider the parameters $\boldsymbol{\theta}$ of a single module in a neural network with $\ell_2$-regularization of strength $\eta$. Let $\boldsymbol{G}(\boldsymbol{\theta}_t)$ denote the curvature matrix and $\nabla_{\boldsymbol{\theta}}\mathcal{L}(\boldsymbol{\theta}_t)$ the gradient at step $t$. One iteration of the optimizer applies

$$\boldsymbol{\theta}_{t+1} \leftarrow \boldsymbol{\theta}_t + \left[\boldsymbol{G}(\boldsymbol{\theta}_t) + (\lambda + \eta)\boldsymbol{I}\right]^{-1}\left[\nabla_{\boldsymbol{\theta}}\mathcal{L}(\boldsymbol{\theta}_t) + \eta\boldsymbol{\theta}_t\right]. \tag{27}$$

The inverse cannot be computed exactly (in reasonable time) for the Kronecker-factored curvatures KFAC, KFLR, and KFRA. We use the scheme of Martens & Grosse (2015) to approximately invert $\boldsymbol{G}(\boldsymbol{\theta}_t) + (\lambda + \eta)\boldsymbol{I}$ if $\boldsymbol{G}(\boldsymbol{\theta}_t)$ is Kronecker-factored; $\boldsymbol{G}(\boldsymbol{\theta}_t) = \boldsymbol{A}(\boldsymbol{\theta}_t) \otimes \boldsymbol{B}(\boldsymbol{\theta}_t)$. It replaces the expression $(\lambda + \theta)\boldsymbol{I}$ by diagonal terms added to each Kronecker factor. In summary, this replaces

$$\left[\boldsymbol{A}(\boldsymbol{\theta}_t) \otimes \boldsymbol{B}(\boldsymbol{\theta}_t) + (\lambda + \eta)\boldsymbol{I}\right]^{-1} \text{ by } \left[\boldsymbol{A}(\boldsymbol{\theta}_t) + \pi_t\sqrt{\lambda + \eta}\boldsymbol{I}\right]^{-1} \otimes \left[\boldsymbol{B}(\boldsymbol{\theta}_t) + \frac{1}{\pi_t}\sqrt{\lambda + \eta}\boldsymbol{I}\right]^{-1}. \tag{28}$$

A principled choice for the parameter $\pi_t$ is given by $\pi_t = \sqrt{\frac{\|\boldsymbol{A}(\boldsymbol{\theta}_t) \otimes \boldsymbol{I}_B\|}{\|\boldsymbol{I}_A \otimes \boldsymbol{B}(\boldsymbol{\theta}_t)\|}}$ for an arbitrary matrix norm $\|\cdot\|$. We follow Martens & Grosse (2015) and choose the trace norm,

$$\pi_t = \sqrt{\frac{\operatorname{tr}(\boldsymbol{A}(\boldsymbol{\theta}_t))\operatorname{dim}(\boldsymbol{B})}{\operatorname{dim}(\boldsymbol{A}) \otimes \operatorname{tr}(\boldsymbol{B}(\boldsymbol{\theta}_t))}}. \tag{29}$$

Table 3: Best hyperparameter settings for optimizers and baselines shown in this work. In the Momentum baselines, the momentum was fixed to 0.9. Parameters for computation of the running averages in Adam use the default values $(\beta_1, \beta_2) = (0.9, 0.999)$. The symbols ✓ and ✗ denote whether the hyperparameter setting is an interior point of the grid or not, respectively.

| Curvature | mnist_logreg | | | fmnist_2c2d | | | cifar10_3c3d | | | cifar100_allcnnc | | |
|---|---|---|---|---|---|---|---|---|---|---|---|---|
| | $\alpha$ | $\lambda$ | int | $\alpha$ | $\lambda$ | int | $\alpha$ | $\lambda$ | int | $\alpha$ | $\lambda$ | int |
| DiagGGN | $10^{-3}$ | $10^{-3}$ | ✓ | $10^{-4}$ | $10^{-4}$ | ✗ | $10^{-3}$ | $10^{-2}$ | ✓ | - | - | - |
| DiagGGN-MC | $10^{-3}$ | $10^{-3}$ | ✓ | $10^{-4}$ | $10^{-4}$ | ✗ | $10^{-3}$ | $10^{-2}$ | ✓ | $10^{-3}$ | $10^{-3}$ | ✓ |
| KFAC | $10^{-2}$ | $10^{-2}$ | ✓ | $10^{-3}$ | $10^{-3}$ | ✓ | 1 | 10 | ✗ | 1 | 1 | ✓ |
| KFLR | $10^{-2}$ | $10^{-2}$ | ✓ | $10^{-2}$ | $10^{-3}$ | ✓ | 1 | 10 | ✗ | - | - | - |
| KFRA | $10^{-2}$ | $10^{-2}$ | ✓ | - | - | - | - | - | - | - | - | - |
| **Baseline** | $\alpha$ | | | $\alpha$ | | | $\alpha$ | | | $\alpha$ | | |
| Momentum | $\approx 2.07 \cdot 10^{-2}$ | | | $\approx 2.07 \cdot 10^{-2}$ | | | $\approx 3.79 \cdot 10^{-3}$ | | | $\approx 4.83 \cdot 10^{-1}$ | | |
| Adam | $\approx 2.98 \cdot 10^{-4}$ | | | $\approx 1.27 \cdot 10^{-4}$ | | | $\approx 2.98 \cdot 10^{-4}$ | | | $\approx 6.95 \cdot 10^{-4}$ | | |

## C.4    ADDITIONAL RESULTS

This section presents the results for MNIST using a logistic regression in Fig. 10 and FASHION-MNIST using the 2C2D network, composed of two convolution and two linear layers, in Fig. 11.

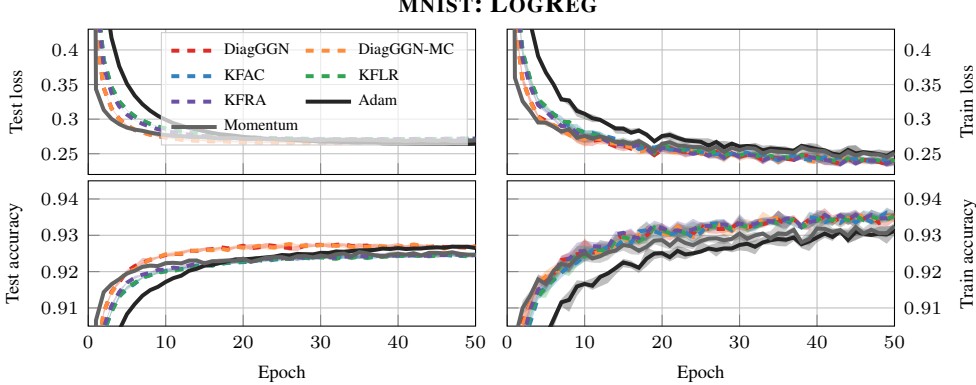

Figure 10: Median performance with shaded quartiles of the best hyperparameter settings chosen by DEEPOBS for logistic regression (7,850 parameters) on MNIST. Solid lines show well-tuned baselines of momentum SGD and Adam that are provided by DEEPOBS.

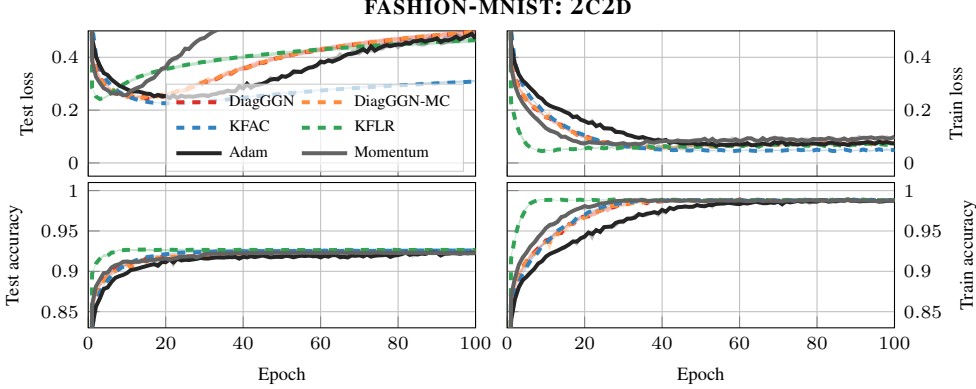

Figure 11: Median performance with shaded quartiles of the best hyperparameter settings chosen by DEEPOBS for the 2C2D network (3,274,634 parameters) on FASHION-MNIST. Solid lines show well-tuned baselines of momentum SGD and Adam that are provided by DEEPOBS.

# D BACKPACK CHEAT SHEET

- Assumptions
  - Feedforward network

$$\boldsymbol{z}_n^{(0)} \xrightarrow{T_{\boldsymbol{\theta}^{(1)}}^{(1)}(\boldsymbol{z}_n^{(0)})} \boldsymbol{z}_n^{(1)} \xrightarrow{T_{\boldsymbol{\theta}^{(2)}}^{(2)}(\boldsymbol{z}_n^{(1)})} \dots \xrightarrow{T_{\boldsymbol{\theta}^{(L)}}^{(L)}(\boldsymbol{z}_n^{(L-1)})} \boldsymbol{z}^{(L)} \xrightarrow{\ell(\boldsymbol{z}_n^{(L)}, \boldsymbol{y})} \ell(\boldsymbol{\theta})$$

  - $d^{(i)}$ : Dimension of parameter $\boldsymbol{\theta}^{(i)}$
  - Empirical risk

$$\mathcal{L}(\boldsymbol{\theta}) = \tfrac{1}{N} \sum_{n=1}^{N} \ell(f(\boldsymbol{\theta}, \boldsymbol{x}_n), \boldsymbol{y}_n)$$

- Shorthands

$$\ell_n(\boldsymbol{\theta}) = \ell(f(\boldsymbol{\theta}, \boldsymbol{x}_n), \boldsymbol{y}_n), \qquad n = 1, \dots, N,$$
$$f_n(\boldsymbol{\theta}) = f(\boldsymbol{\theta}, \boldsymbol{x}_n) = \boldsymbol{z}_n^{(L)}(\boldsymbol{\theta}), \qquad n = 1, \dots, N$$

- Generalized Gauss-Newton matrix

$$\boldsymbol{G}(\boldsymbol{\theta}) = \frac{1}{N} \sum_{n=1}^{N} (\mathrm{J}_{\boldsymbol{\theta}} f_n)^{\top} \nabla_{f_n}^2 \ell_n(\boldsymbol{\theta}) (\mathrm{J}_{\boldsymbol{\theta}} f_n)$$

- Approximative GGN via MC sampling

$$\tilde{\boldsymbol{G}}(\boldsymbol{\theta}) = \frac{1}{N} \sum_{n=1}^{N} (\mathrm{J}_{\boldsymbol{\theta}} f_n)^{\top} \left[ \nabla_{\boldsymbol{\theta}} \ell(f_n(\boldsymbol{\theta}), \hat{\boldsymbol{y}}) \nabla_{\boldsymbol{\theta}} \ell(f_n(\boldsymbol{\theta}), \hat{\boldsymbol{y}}_n)^{\top} \right]_{\hat{y}_n \sim p_{f_n(\boldsymbol{x}_n)}} (\mathrm{J}_{\boldsymbol{\theta}} f_n)$$

Table 4: Overview of the features supported in the first release of BACKPACK. The quantities are computed separately for all module parameters, i.e. $i = 1, \dots, L$.

| Feature | Details |
|---|---|
| Individual gradients | $\frac{1}{N} \nabla_{\boldsymbol{\theta}^{(i)}} \ell_n(\boldsymbol{\theta}), \quad n = 1, \dots, N$ |
| Batch variance | $\frac{1}{N} \sum_{n=1}^{N} [\nabla_{\boldsymbol{\theta}^{(i)}} \ell_n(\boldsymbol{\theta})]_j^2 - [\nabla_{\boldsymbol{\theta}^{(i)}} \mathcal{L}(\boldsymbol{\theta})]_j^2, \qquad j = 1, \dots, d^{(i)}$ |
| 2nd moment | $\frac{1}{N} \sum_{n=1}^{N} [\nabla_{\boldsymbol{\theta}^{(i)}} \ell_n(\boldsymbol{\theta})]_j^2, \quad j = 1, \dots, d^{(i)}.$ |
| Indiv. gradient $\ell_2$ norm | $\left\| \frac{1}{N} \nabla_{\boldsymbol{\theta}^{(i)}} \ell_n(\boldsymbol{\theta}) \right\|_2^2, \quad n = 1, \dots, N$ |
| DiagGGN | $\mathrm{diag}\left( \boldsymbol{G}(\boldsymbol{\theta}^{(i)}) \right)$ |
| DiagGGN-MC | $\mathrm{diag}\left( \tilde{\boldsymbol{G}}(\boldsymbol{\theta}^{(i)}) \right)$ |
| Hessian diagonal | $\mathrm{diag}\left( \nabla_{\boldsymbol{\theta}^{(i)}}^2 \mathcal{L}(\boldsymbol{\theta}) \right)$ |
| KFAC | $\tilde{\boldsymbol{G}}(\boldsymbol{\theta}^{(i)}) \approx \boldsymbol{A}^{(i)} \otimes \boldsymbol{B}_{\mathrm{KFAC}}^{(i)}$ |
| KFLR | $\boldsymbol{G}(\boldsymbol{\theta}^{(i)}) \approx \boldsymbol{A}^{(i)} \otimes \boldsymbol{B}_{\mathrm{KFLR}}^{(i)}$ |
| KFRA | $\boldsymbol{G}(\boldsymbol{\theta}^{(i)}) \approx \boldsymbol{A}^{(i)} \otimes \boldsymbol{B}_{\mathrm{KFRA}}^{(i)}$ |

