# OpenReview forum: "BackPACK: Packing more into Backprop"
_ICLR.cc/2020/Conference — Accept (Talk)_

### Official Review · AnonReviewer1 · 2019-10-27
**Official Blind Review #1**

**Rating:** 8

**Review:**

This paper adds a very interesting and useful feature to existing autodifferentiation for training neural networks. The second-order information can be backprogated just as the first-order ones, which can be used to accelerate training.

This idea, although according to the paper, is developed upon existing works, still, strongly attracts as the second-order information is crucial for training and perhaps visualizing the landscape of neural networks. I vote for an acceptance as this brings a significantly important feature to PyTorch, and the author's good experiments results and open-sourced code.

**Experience Assessment:**

I have published one or two papers in this area.

**Review Assessment: Checking Correctness Of Derivations And Theory:**

I assessed the sensibility of the derivations and theory.

**Review Assessment: Checking Correctness Of Experiments:**

I assessed the sensibility of the experiments.

**Review Assessment: Thoroughness In Paper Reading:**

I read the paper at least twice and used my best judgement in assessing the paper.

---

> ### Author Response · Authors · 2019-11-11
> **Re: Official Blind Review #1**
>
> Many thanks for your strong review, we’re glad to hear you are pleased with the paper! Thanks for arguing in our favor!

---

### Official Review · AnonReviewer3 · 2019-10-28
**Official Blind Review #4**

**Rating:** 8

**Review:**

This is a good paper. The authors present a software implementation which allows one to extend PyTorch to compute quantities that are irritatingly difficult to compute in PyTorch directly, or in other automatic differentiation frameworks, particularly if efficiency is a concern. Issues of this kind have been discussed at length within the community, particularly on GitHub, and related issues with optimization of automatic differentiation code have motivated other software developments, such as Julia's Zygote package. Having wasted a large amount of my time implementing the WGAN gradient penalty from scratch - which, to implement scalably, requires one to use both forward-mode and reverse-mode automatic differentiation simultaneously - I appreciate what the authors are trying to do here to make research that requires more sophisticated automatic differentiation more accessible.

---

Detailed remarks below.

* The paper's title is not very informative about what the paper is about. The authors should choose a different title - perhaps something like "BackPACK: user-friendly higher-order automatic differentiation in deep networks" or something similar but less long.

* The authors focus on KFAC-type methods as their key illustrated use case, but actually software frameworks like this are also helpful for certain GAN losses - WGAN-GP in particular. These losses require one to compute a term that involves the norm of a certain gradient of the output. The gradient of this gradient can be handled efficiently with Hessian-Vector products, which in principle are computable efficiently via automatic differentiation, but in practice a huge pain because of the need to use both forward-mode and reverse-mode automatic differentiation and lack of first-class support from automatic differentiation packages. Provided I've not misunderstood BackPACK and that it would help in making such computations less tedious (and I can't verify this myself, as my own implementation of WGAN-GP was not in PyTorch), I would highly recommend the authors to add an extra paragraph discussing this particular use case, because this would increase the paper's impact on the community by connecting it to another literature which is not mentioned in the paper.

* The entire paper could do with talking about backpropagation less, and automatic differentiation more, because it illustrates that the concerns addressed are not solely limited to deep networks, even if the package does primarily target them.

* P2: these issues are not limited to the Python community, and specialized automatic differentiation software has also been developed for Julia. The authors should cite Innes [1,2] and related papers from that literature.

* Figure 1: from the sample code, I worry about how generic BackPACK is. I think the package authors should be careful not to specialize too much to particular kinds of deep networks, particularly since a much wider variety of models and network architectures are starting to be used with automatic differentiation.

* P2: capital \Omega notation is confusing, please replace with capital \Theta.

* P2: L_2 regularization should more technically be \ell^2 instead.

* P4: please cite Baydin [3] who provides a very nice review of automatic differentiation. It may help explanation to introduce dual numbers, which make forward-mode much easier to understand.

* P6: please write out "with respect to" for "w.r.t.".

* P7: I really liked this section. The simplicity of implementing the example method using the authors' software framework feels compelling to me. However, genericness is still a concern: by analogy, every deep learning framework can do MNIST easily, but some of them make it much harder to do customized or advanced implementation than others. The latter cases are often the ones that matter to practitioners. It's hard to tell how easy it will be to implement something the authors did not foresee or consider - but this will necessarily be the case in any software paper.

* P8: "and in part driven" - missing a comma.

* P8: please spell out "Table" in "Tab. 1".

[1] M. Innes. Don't Unroll Adjoint: Differentiating SSA-Form Programs. NeurIPS, 2018.
[2] M. Innes. Flux: Elegant Machine Learning with Julia. Journal of Open Source Software, 2018.
[3] Baydin, Atilim Gunes and Pearlmutter, Barak A and Radul, Alexey Andreyevich and Siskind, Jeffrey Mark. Automatic differentiation in machine learning: a survey. JMLR, 2017.

**Experience Assessment:**

I have read many papers in this area.

**Review Assessment: Checking Correctness Of Derivations And Theory:**

N/A

**Review Assessment: Checking Correctness Of Experiments:**

I assessed the sensibility of the experiments.

**Review Assessment: Thoroughness In Paper Reading:**

I read the paper thoroughly.

---

> ### Author Response · Authors · 2019-11-11
> **Re: Official Blind Review #4**
>
> Thanks for your detailed reading, all typographic and related work remarks will of course be addressed! On your two specific comments:
>
> We appreciate your comment on the title and will try to find a more descriptive one.
> However, like you, we so far have struggled to find a sufficiently compact one.
>
> And thanks for your comments on WGAN-GP, we sympathize with your frustration. We will add a discussion of Hessian-vector products as they are necessary for modern models and algorithms, and have limited support in deep learning frameworks.
>
> BackPACK provides ease-of-use functions for Hessian-vector products in PyTorch, which use backward-on-backward [1,2], to make them more user-friendly and broaden the usability of this functionality. Unfortunately, the implementation of forward-on-backward requires more integration into PyTorch than is reasonably possible as a third party.
>
> Our core contribution remains the computation of side-products of the backward pass in PyTorch. To minimize overhead, BackPACK piggybacks on top of the existing backward pass to extract quantities that are not the direct result of autodifferentiation, for example the variance of the gradient or KFAC.
>
> [1] Fast exact multiplication by the Hessian
> Pearlmutter, 1993
> [2] Fast curvature matrix-vector products for second-order gradient descent
> Schraudolph, 2002

---

### Official Review · AnonReviewer2 · 2019-10-29
**Official Blind Review #2**

**Rating:** 8

**Review:**

This paper presents a Pytorch framework for experimenting with first and second order extensions to standard gradient updates via backpropagation.  At the time of writing, the implementation supports feed-forward networks where there is a strict module ordering (by which I mean residual connections are not possible).

The framework enables researchers to easily experiment with techniques which involve modifying the update according to quantities computed over the batch of gradients (i.e. before they are summed to form the standard SGD update)—these are ‘first-order’ extensions—and it also makes use of the block-diagonal factorisation of the Hessian outlined in Mizutani & Dreyfus as well as Dangel & Hennig to enable the computation of second order quantities via ~ ‘hessian prop’.

I think the paper is a strong accept: the framework has some limitations in the current form (mostly in terms of what architectures are supported), however it still provides a very useful and extensible tool for researchers to efficiently experiment with a variety of more complex optimisation architectures.  This is (as the paper states) a large bottleneck for much optimisation research in deep learning.

In section 2.3 you state that generalised Gauss-Newton (GGN) is guaranteed positive semi-definite.  It would also be nice to add a sentence as to when (even intuitively) the Fisher information coincides with GGN; (in practice, as the GGN uses a (possibly rank-bounded) sample size of ‘N’, while the Fisher is the expectation under the data generating distribution, one could argue that even when they should be ==, it would only be as N->\infty).





**Experience Assessment:**

I have read many papers in this area.

**Review Assessment: Checking Correctness Of Derivations And Theory:**

I assessed the sensibility of the derivations and theory.

**Review Assessment: Checking Correctness Of Experiments:**

I carefully checked the experiments.

**Review Assessment: Thoroughness In Paper Reading:**

I read the paper thoroughly.

---

> ### Author Response · Authors · 2019-11-11
> **Re: Official Blind Review #2**
>
> Many thanks for your feedback, we are glad you are pleased with the paper. We agree with your assessment and are working to address the limitations. The support for more architectures such as recurrent and residual layers is on the drawing board for V2.
>
> We will make more clear in §2.3 that the GGN is equivalent to the (empirical) Fisher information matrix used in Amari’s natural gradient descent [1,2,3]. Those connections are explored in more detail in §9 of Martens [4]. The connection to the “full” Fisher information matrix that uses the distribution over the input data is trickier, but should hold as N → ∞.
>
> [1]: Natural gradient works efficiently in learning
> Amari 1998
> [2]: Adaptive natural gradient learning algorithms for various stochastic models
> Park, Amari, Fukumizu, 2000
> [3]: Universal Statistics of Fisher Information in Deep Neural Networks: Mean Field Approach
> Karakida, Akaho, Amari, 2018
> [4]: New insights and perspectives on the natural gradient method
> Martens, 2014

---

### Decision · Program_Chairs · 2019-12-19

**Decision:**

Accept (Talk)

**Comment:**

The paper efficiently computes quantities, such as variance estimates of the gradient or various Hessian approximations, jointly with the gradient, and the paper also provides a software package for this. All reviewers agree that this is a very good paper and should be accepted.